# Reactive oxygen species-responsive and Raman-traceable hydrogel combining photodynamic and immune therapy for postsurgical cancer treatment

Yiyi Zhang[1,7], Sidan Tian[1,7], Liping Huang[1,7], Yanan Li[2,3,7], Yuan Lu[1], Hongyu Li[1], Guiping Chen[4], Fanling Meng[1,5], Gang L. Liu[1], Xiangliang Yang [1,5,6], Jiasheng Tu[2], Chunmeng Sun[2] & Liang Luo [1,5,6] ✉

Combining immune checkpoint blockade (ICB) therapy with photodynamic therapy (PDT) holds great potential in treating immunologically "cold" tumors, but photo-generated reactive oxygen species (ROS) can inevitably damage co-administered ICB antibodies, hence hampering the therapeutic outcome. Here we create a ROS-responsive hydrogel to realize the sustained co-delivery of photosensitizers and ICB antibodies. During PDT, the hydrogel skeleton poly(deca-4,6-diynedioic acid) (PDDA) protects ICB antibodies by scavenging the harmful ROS, and at the same time, triggers the gradual degradation of the hydrogel to release the drugs in a controlled manner. More interestingly, we can visualize the ROS-responsive hydrogel degradation by Raman imaging, given the ultrastrong and degradation-correlative Raman signal of PDDA in the cellular silent window. A single administration of the hydrogel not only completely inhibits the long-term postoperative recurrence and metastasis of 4T1-tumor-bearing mice, but also effectively restrains the growth of re-challenged tumors. The PDDA-based ROS-responsive hydrogel herein paves a promising way for the durable synergy of PDT and ICB therapy.

Immune checkpoint blockade (ICB) therapy has emerged as a mainstay of the first-line modalities in the clinical treatment of diverse malignances[1-4], but it undertakes a major challenge that only a small subset of patients receive clinical benefits. Many solid tumors, which are considered as immunologically "cold"[5-8], respond poorly to ICB therapies because of their insufficient tumor-infiltrating lymphocytes (TILs) and rich tumor-promoting macrophages, regulatory T cells (Tregs), and anti-inflammatory cytokines[9-12]. Combining ICB therapy with photodynamic therapy (PDT) denotes a promising strategy to potentiate the immune systems for "cold" tumor treatment. The reactive oxygen species (ROS) generated in PDT not only kills tumor cells directly, but also stimulates the pro-inflammatory effects and

[1]National Engineering Research Center for Nanomedicine, College of Life Science and Technology, Huazhong University of Science and Technology, Wuhan 430074, China. [2]NMPA Key Laboratory for Research and Evaluation of Pharmaceutical Preparations and Excipients, Department of Pharmaceutics, School of Pharmacy, China Pharmaceutical University, 24 Tong Jia Xiang, Nanjing 210009, China. [3]School of Food Science and Pharmaceutical Engineering, Nanjing Normal University, Nanjing 210023, China. [4]Bruker (Beijing) Scientific Technology Company Limited, Shanghai Branch, Shanghai 200233, China. [5]Key Laboratory of Molecular Biophysics of the Ministry of Education, College of Life Science and Technology, Huazhong University of Science and Technology, Wuhan 430074, China. [6]Hubei Key Laboratory of Bioinorganic Chemistry and Materia Medica, School of Chemistry and Chemical Engineering, Huazhong University of Science and Technology, Wuhan 430074, China. [7]These authors contributed equally: Yiyi Zhang, Sidan Tian, Liping Huang, Yanan Li. ✉e-mail: liangluo@hust.edu.cn

consequently enhances dendritic cells (DCs) maturation and TIL recruitment[13–15]. More importantly, ROS-related redox signaling plays a key role in transforming macrophages from tumor-promoting M2 phenotype to tumor-inhibiting M1 phenotype[16,17]. However, excessive ROS also reacts with adjacent antibodies[18–20]. Especially, when photosensitizers (PSs) are co-delivered with ICB antibodies in confined carriers, the generated ROS inside the carrier can inevitably destruct unreleased antibodies. The therapeutic outcomes of combined PDT and ICB therapy is therefore significantly impeded. An ideal sustainable co-delivering systems of PSs and ICB antibodies, which can preserve the activity of antibodies and scavenge the damaging ROS, are greatly desired for efficient synergy of cancer immunotherapy and PDT.

In this work, we have created a biocompatible, ROS-responsive hydrogel PPG, based on the crosslinking of a conjugated polymer poly(deca-4,6-diynedioic acid) (PDDA)[21] and a natural polysaccharide pullulan[22,23], to approach such an ideal co-delivery of ICB antibodies and PSs (Fig. 1a). We have reported previously that PDDA completely degrades into biocompatible succinic acid upon sunlight irradiation in air[21]. Here we have further discovered that this environmentally degradable polymer can react extensively with ROS generated by PS such as Chlorin e6 (Ce6), and decompose completely into succinic acid as well. We hypothesize that the PPG, which utilizes PDDA as the scaffold skeletons, can scavenge the harmful ROS generated during PDT to protect antibodies, and at the same time, trigger the gradual degradation of PPG through this oxidative decomposition to release drugs into tumor microenvironment (TME) in a controlled manner. There have been many deliberately designed ROS-responsive hydrogels[24–26], but PPG

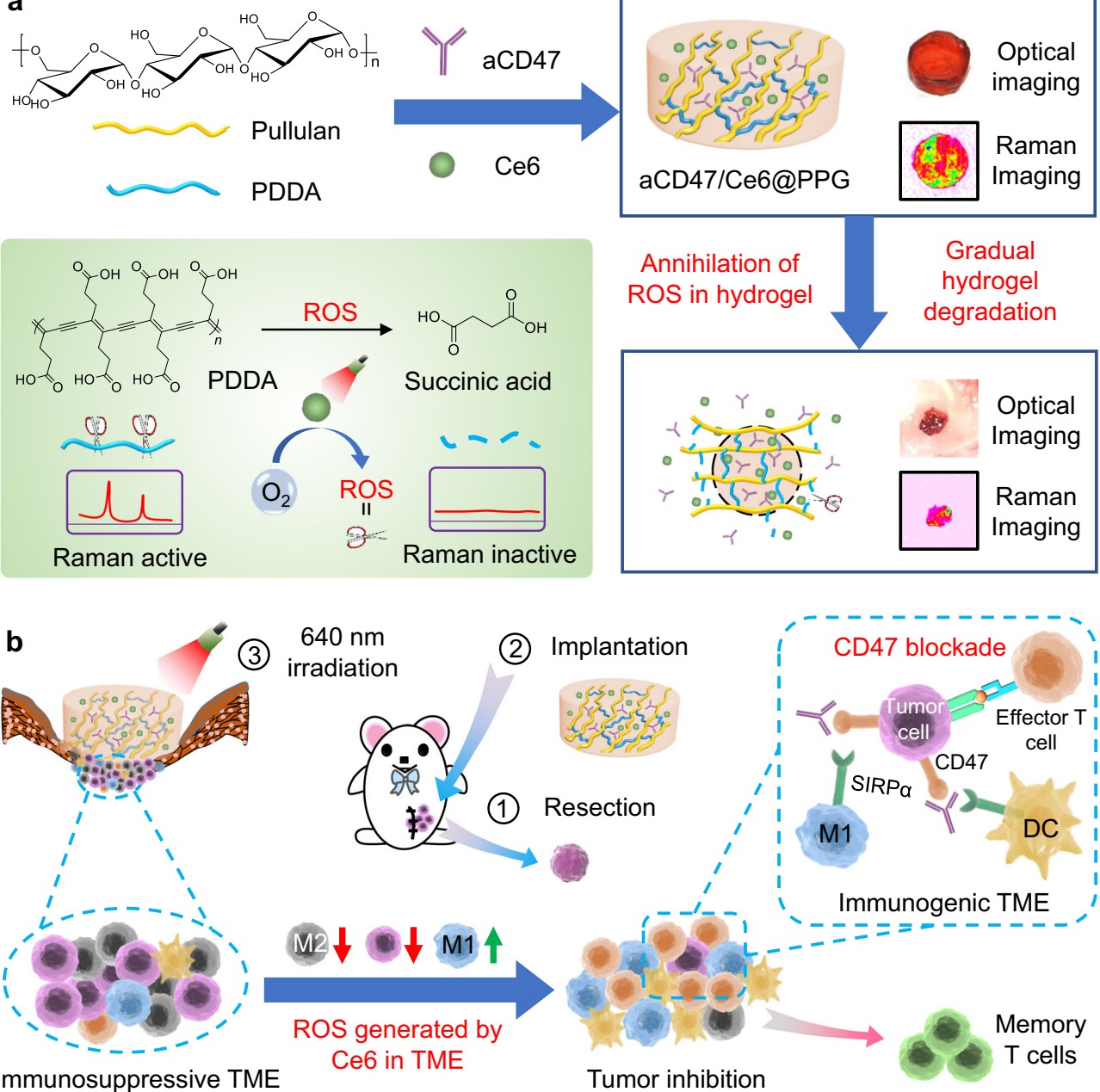

**Fig. 1 | Schematic illustration of the ROS-responsive hydrogel for the sustained co-delivery of photosensitizers and ICB antibodies. a** Schematic illustration of the construction of aCD47/Ce6@PPG hydrogel and its Raman-traceable, ROS-responsive degradation. **b** Synergistic combination of PDT and aCD47 therapy on postsurgical 4T1 models enabled by the ROS-responsive degradation of the hydrogel.

undergoes a completely different ROS-scavenging mechanism and yields biocompatible succinic acid after degradation. More strikingly, PDDA has an ultrastrong Raman signal (2121 cm$^{-1}$) in the cellular silent window (1800–2800 cm$^{-1}$), which diminishes accordingly with its ROS-responsive degradation. The straight correlation between the Raman signal of PDDA and its backbone structure warrants that the degradation of the hydrogel can be precisely traced by the Raman signal change of PDDA[27,28]. The fate of drug carriers in vivo remains commonly undiscovered, and loading imaging agents cannot project the intrinsic degradation behavior of the carriers. Raman imaging based on the scaffold-forming PDDA should enable the visualization of its ROS-responsive degradation process by Raman imaging, which is greatly favorable for the direct justification of the consumption of harmful ROS as well as for the refined control of degradation-dependent drug release.

We have validated this ROS-responsive, Raman-traceable hydrogel in the postsurgical treatment of 4T1 tumor model, a typical immune-cold tumor with scarce TILs[12] (Fig. 1b). Breast cancer has the leading incidence and mortality rates among females[29], and surgery is still the most effective treatment regimen, whereas the long-term prognosis is unsatisfactory due to local recurrence and metastasis. ICB therapy is especially unsuccessful in postsurgical treatment of breast cancer, attributed to its highly immunosuppressive TME. In addition, the overexpression of the integrin-associated protein (CD47) on the surface of tumor cells, which can send the antiphagocytic "don't eat me" signal to macrophages and evade phagocytosis, is another crucial factor of tumor invasion[30–32]. Blocking the CD47-SIRPα pathway by anti-CD47 antibody (aCD47) can activate macrophages and other important antigen presenting cells to enhance the antitumor response mediated by TILs.

Here, we load aCD47 and Ce6 in the hydrogel and apply it on the postsurgical 4T1 mouse models. Strikingly, we achieve successful long-term inhibition of tumor recurrence and metastasis on 4T1 models by a single administration of the aCD47/Ce6@PPG hydrogel, which demonstrates that this immunostimulant combination, enabled by our ROS-responsive, Raman-traceable hydrogel, holds great potential as a safe and universal therapeutic platform for the treatment of immunologically "cold" tumors in clinic.

## Results

### Complete degradation of PDDA by Ce6

The complete degradation of PDDA by PSs upon appropriate light irradiation forms the basis of ROS-responsive PPG hydrogel. As a conjugated polymer, PDDA is stable in the dark or hypoxia conditions. However, when exposed to sunlight and air, PDDA disintegrates rapidly and fully decomposes through photooxidation within a few days, yielding biocompatible succinic acid as a major degradation product[21]. We are hereby inspired to further investigate whether it can completely degrade under other controlled oxidative conditions. We dispersed PDDA into a Ce6-containing aqueous solution and applied a red light (640 nm, 5 mW cm$^{-2}$) on the mixture (Fig. 2a). The absorption peak associated to PDDA (460 nm) decreased with the increase of irradiation time (Fig. 2b), suggesting the gradual depletion of its conjugated backbone. The characteristic Raman peaks corresponding to the C=C bond (1522 cm$^{-1}$) and C≡C bond (2121 cm$^{-1}$) in the PDDA backbone also diminished accordingly with elongated irradiation time (Fig. 2c). Strikingly, the absorbance at 460 nm ($A_{460\,nm}$) and the two Raman peaks (Raman $I_{1522\,cm^{-1}}$ and Raman $I_{2121\,cm^{-1}}$) decreased synchronously (Fig. 2d), which clearly evidenced the concurrent and thorough depletion of the polymer backbone of PDDA under photodynamic conditions. Further examining the degradants of PDDA using NMR and mass spectroscopy (Fig. 2e–g) evidenced that succinic acid was the dominant degradation product.

### Design, preparation, and characterization of PPGs

Affirmed by the complete degradation of PDDA using photo-irradiated Ce6, we then synthesized PPG by crosslinking pullulan with PDDA through the condensation between the carboxyl groups on PDDA and the hydroxyl groups on pullulan (Fig. 3a), catalyzed by N-(3-dimethylaminopropyl)-N'-ethylcarbodiimide hydrochloride (EDC) and 4-dimethylaminopyridine (DMAP). To load drugs into the hydrogel, PPG was lyophilized and the stock solutions of aCD47 and Ce6 were added to the dry PPG until they were completely absorbed (Supplementary Fig. 1). Interestingly, PPGs with various crosslinking densities could be achieved by simply tuning the PDDA/pullulan mass ratios, in a range from 1:4 to 1:20. Scanning electron microscopy (SEM) images revealed the uniform porous features of these PPGs, and the pore size in these PPGs increased accordingly with the decrease of the amount of PDDA (Fig. 3b).

All the prepared PPGs exhibited good swelling properties to allow for sufficient drug loading in them (Fig. 3c). On the other hand, the mechanical strength of the PPGs with PDDA/pullulan mass ratios of 1:15 and 1:20 was relatively weaker (Fig. 3d and Supplementary Fig. 2), attributed to their lower crosslinking densities. As expected, the degradation of the prepared PPG accelerated in the presence of increased ROS levels (Supplementary Fig. 3). Moreover, the in vivo degradation of blank PPGs with various PDDA/pullulan mass ratios showed that with the increase of the PDDA/pullulan mass ratio, the time for the complete degradation of PPG after being implanted in vivo extended gradually (Supplementary Fig. 4). With 1:15 and 1:20 PDDA/pullulan mass ratios, the PPGs completely degraded after being implanted for less than 20 days. As a natural polysaccharide with good biocompatibility and biodegradability[23,24], pullulan can degrade in physiological conditions. Therefore, both the oxidative cleavage of PDDA and the metabolic breakage of pullulan would contribute to the degradation the hydrogel and the consequent drug release. Considering that photo-generated ROS would further accelerate the degradation of hydrogel in PDT, the PPGs with 1:15 and 1:20 PDDA/pullulan mass ratios were unsuitable for the desired sustainable release of aCD47[33,34]. When the PDDA/pullulan mass ratios increased to 1:8, the implanted PPG degraded significantly slower. However, further increasing the mass ratios of PDDA/pullulan (1:6 and 1:4) led to negligible changes of the implanted PPGs on day 20. Collectively, PPG with 1:8 PDDA/pullulan mass ratio was deemed a good carrier candidate, given its appropriate balance in swelling ability, mechanical properties, and in vivo degradation behavior.

### ROS-responsive scaffold degradation and drug release

We next loaded Ce6 into the PPG with a PDDA/pullulan mass ratio of 1:8 to assess the hydrogel degradation driven by ROS generated inside the carrier in vivo. 4T1-tumor-bearing mice were randomly divided into two groups, and the tumors were dissected when they reached ~300 mm$^3$. Each mouse was implanted with Ce6@PPG at the dissection site before suture. One group of the mice were irradiated by a LED (640 nm, 5 mW cm$^{-2}$, 20 min) on the dissection site on day 0, 2, 4, and 6 after the surgery, whereas the other group received no light irradiation. The morphological changes of all implanted Ce6@PPGs were monitored in the following 28 days (Fig. 4a). For the mice receiving light irradiation, we observed significant decrease in the size of implanted Ce6@PPGs on day 14 after implantation, and all the samples completely degraded within 21 days. As a comparison, for the mice without light irradiation, the degradation of the implanted Ce6@PPGs was much slower, where only slight size reduction was observed on day 14, and completely degradation could not be observed until day 28.

Notably, we have also obtained the Raman images of the hydrogel by harvesting the 2120 cm$^{-1}$ Raman signals in the same observation area. The in vivo application of Raman imaging is usually difficult, because of the interference of tissue background and the weak signals

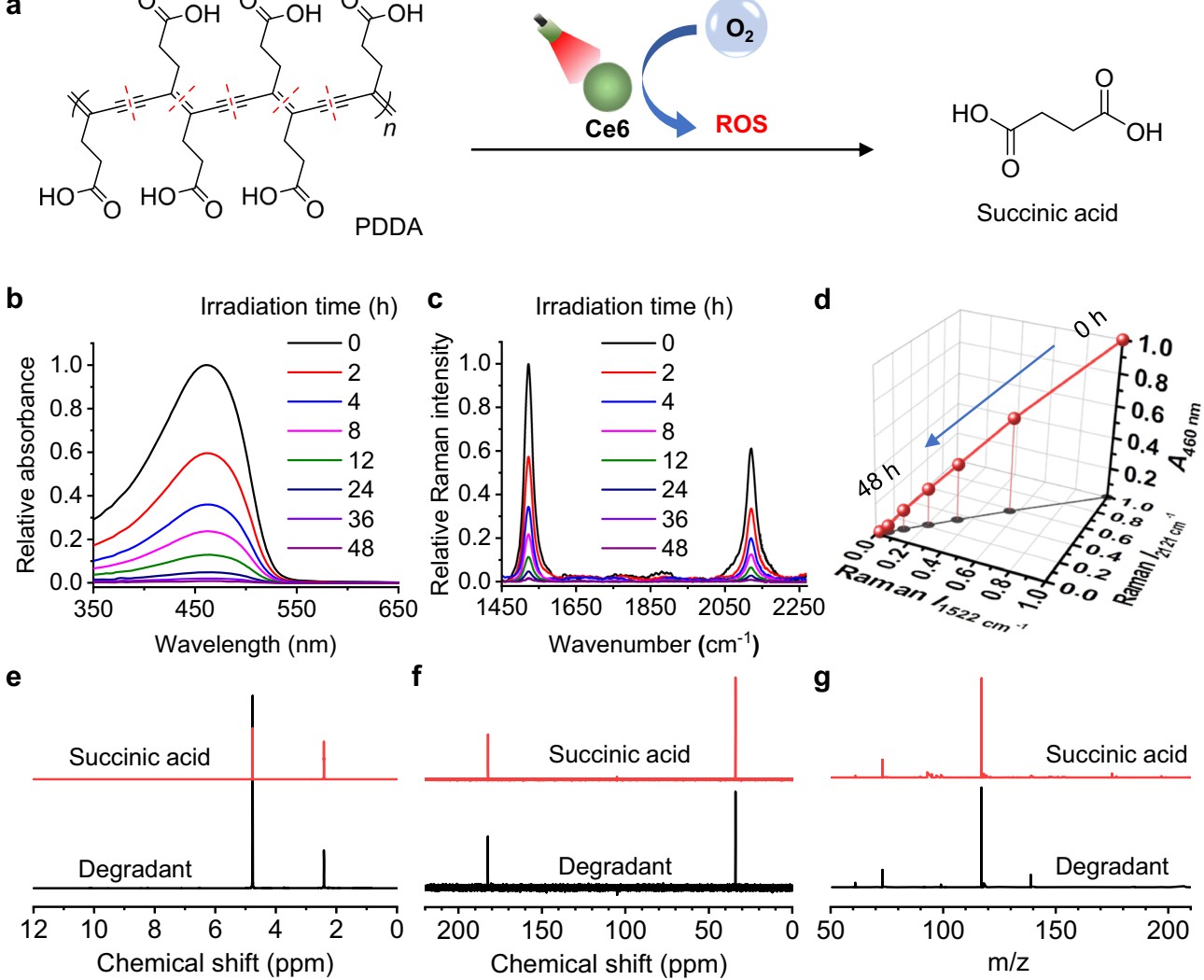

**Fig. 2 | Complete degradation of PDDA by Ce6 upon red light irradiation.**
**a** Degradation reaction of PDDA in the presence of Ce6 with the irradiation.
**b** Absorption spectra of the mixed solution of PDDA and Ce6 after exposure to the irradiation for different time. **c** Raman spectra of the mixed solution of PDDA and Ce6 after exposure to the irradiation for different time. **d** Changes in the relative intensities of PDDA absorbance at 460 nm ($A_{460\,nm}$, Raman at 1522 cm$^{-1}$ (Raman $I_{1522\,cm^{-1}}$) and 2121 cm$^{-1}$ (Raman $I_{2121\,cm^{-1}}$) as a function of irradiation time. $^1$H NMR (**e**), $^{13}$C NMR spectra (**f**), and HR-MS spectra (**g**) of PDDA degradants, in comparison with a succinic acid standard. The red-light irradiation: 640 nm, 5 mW cm$^{-2}$.

of the probes[35–38]. The Raman signal of 2120 cm$^{-1}$ corresponds to the carbon-carbon triple bond stretching (C≡C) in PDDA backbones, which is right in the biological Raman silence window (1800–2800 cm$^{-1}$), so that the Raman images were free of tissue interference (Fig. 4a). In addition, the ultrastrong Raman signals of PDDA, attributed to its conjugated backbone and high polarizability, also enable rapid Raman imaging of the hydrogel at low Raman laser intensity[28]. The change of the Raman signal at 2120 cm$^{-1}$ justified the consumption of damaging ROS generated by Ce6 inside the hydrogel. We next evaluated whether Raman imaging was able to track the degradation of the hydrogel. There were two major degradation-inducing factors of the hydrogel, the oxidative cleavage of PDDA and the metabolic breakage of pullulan. In our study, the Raman image of each Ce6@PPG sample at every time point during the degradation process showed an identical morphology to the corresponding optical image (Fig. 4a), suggesting that the Raman imaging could be used to track the morphological change of implanted hydrogels during the degradation in real time.

To examine whether the ROS-responsive degradation of PPG could be used for the sustained and controlled release of antibodies, we prepared the IgG/Ce6@PPG hydrogel, in which IgG was used as a model antibody to be coloaded with Ce6 in PPG. As shown in Supplementary Fig. 5, without 640 nm LED irradiation, only 6% of IgG could escape from the hydrogel after 15 days. However, when light irradiation was applied (20 min daily on day 0, 2, 4, and 6), significantly enhanced release of IgG was observed, attributed to the irradiation-boosted ROS production by Ce6 and the consequently promoted degradation of PPG. To visualize the ROS-responsive release behavior in vivo, AF790-labeled goat anti-mouse IgG (AFIgG) was used to form the AFIgG/Ce6@PPG hydrogels, which were next implanted into the tumor resection cavities of mice in two random groups. Compared with the mice injected with free Ce6 or AFIgG solutions, the fluorescence signals of Ce6 (Fig. 4b, c) and AFIgG (Fig. 4d, e) in the mice implanted with AFIgG/Ce6@PPG attenuated considerably slower, observed using an in vivo imaging system (IVIS). Irradiation could accelerate the drug release of both Ce6 and AFIgG, which validated that the consumption of ROS generated inside the hydrogel by PDDA could tune the hydrogel degradation for controlled and sustainable drug release.

## Protecting aCD47 during PDT
We also evaluated the protection effect of PDDA on aCD47 in the presence of Ce6-generated ROS. We first used bovine serum albumin

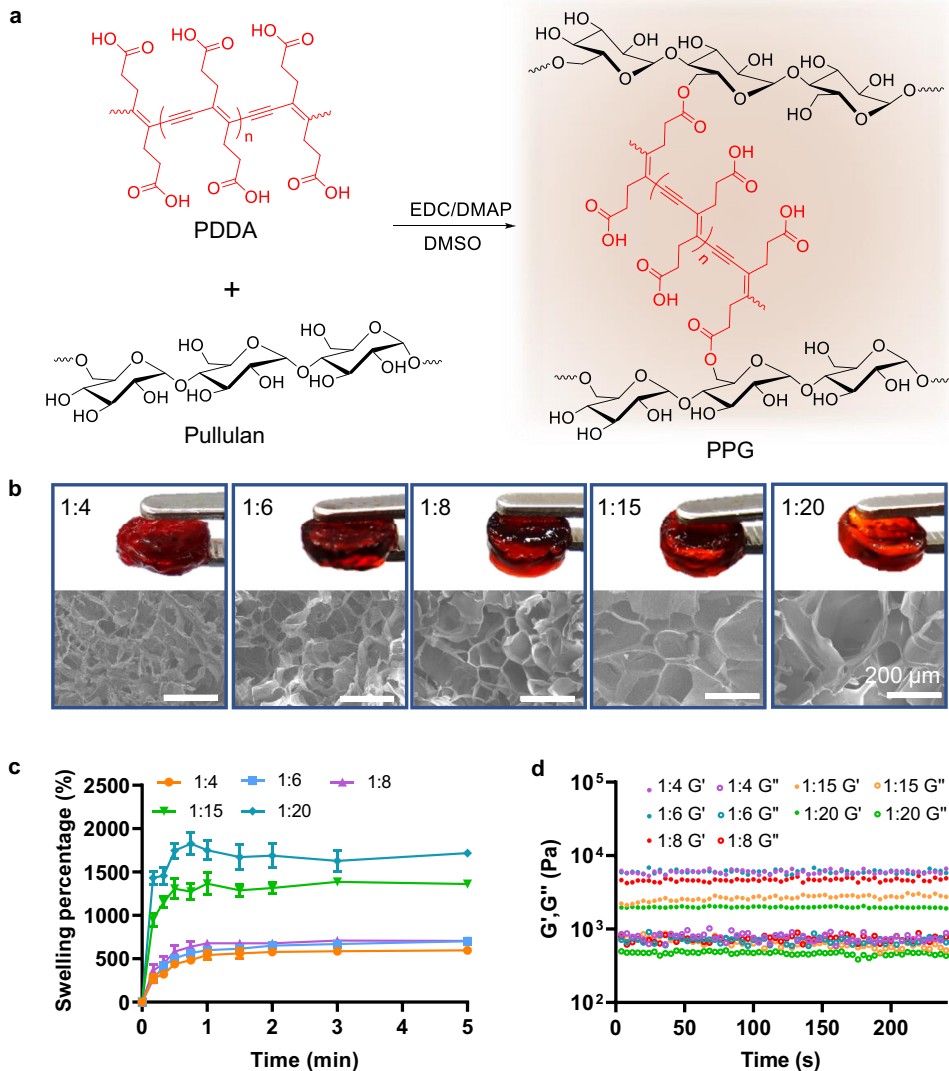

**Fig. 3 | Preparation and characterization of PPGs. a** Crosslinking between PDDA and pullulan through EDC/DMAP catalyzed condensation. **b** Optical and SEM images of PPGs formed with different PDDA/pullulan mass ratio. Scale bar: 200 μm. **c** Swelling behaviors of lyophilized PPGs. Data are shown as mean ± SEM ($n = 3$ independent samples). **d** Time sweep rheological tests of as prepared PPGs.

(BSA) as a model protein, which could be denatured by oxidation. When co-incubated with Ce6, the band of BSA on gel electrophoresis was significantly weakened upon 640 nm light irradiation (Supplementary Fig. 6). Interestingly, in the presence of PDDA, a large proportion of the protein could be rescued, suggesting that PDDA was effective in protecting BSA from ROS-triggered damage. We next investigated whether the ROS consumption by PDDA could preserve the activity of aCD47 using Nano Surface Plasmon Resonance (NanoSPR)[39]. As shown in Supplementary Fig. 7, when the mixed solution of aCD47 and Ce6 was irradiated by LED light (640 nm, 5 mW cm$^{-2}$), the binding affinity between aCD47 and CD47 antigen decreased obviously, suggesting that Ce6-generated ROS could inhibit the activity of aCD47. Interestingly, in the presence of PDDA, light irradiation did not affect the binding affinity between aCD47 and CD47 antigen. To further examine the antibody protective effect of PPG hydrogel, we also used NanoSPR to investigate the change of binding affinity of aCD47 antibody and CD47 antigen in the presence and absence of PPG. Without PPG hydrogel, the antibody-antigen dissociation rate ($K_d$) of free aCD47 greatly increased to $3.04 \times 10^{-4}$ s$^{-1}$ and the antibody-antigen effective binding concentration $K_D$ increased to

$4.59 \times 10^{-9}$ M when Ce6 and light irradiation were applied, demonstrating the deactivation effect of Ce6-generated ROS on aCD47. As a comparison, in the presence of PPG, the value of $K_d$ decreased to $5.79 \times 10^{-5}$ s$^{-1}$ and $K_D$ decreased to $2.51 \times 10^{-9}$ M (Supplementary Table 1), which clearly evidenced that PDDA was able to preserve the activity of aCD47 by consuming the Ce6-generated ROS.

## In vivo antitumor efficacy of aCD47/Ce6@PPG hydrogel
To evaluate the antitumor efficacy of aCD47/Ce6@PPG hydrogel, we used an incompletely resected 4T1-tumor mouse model and investigated the suppression of tumor recurrence after surgery (Fig. 5a). In order to mimic the situation of incomplete tumor removal in clinic, 4T1-luc breast cancer cells were inoculated into the flanks of mice, and ~90% of the tumor was surgically resected when the tumor volume reached ~300 mm$^3$ (Fig. 5b). The mice were randomly grouped, which were respectively treated with PBS solution, a mixed solution of aCD47 and Ce6 (free aCD47/Ce6), blank PPG, Ce6@PPG, or aCD47/Ce6@PPG on the resection site before suture (Fig. 5b). The dose administered on each mouse was set as 80 μg for Ce6 and 70 μg for aCD47. The surgical beds on the mice

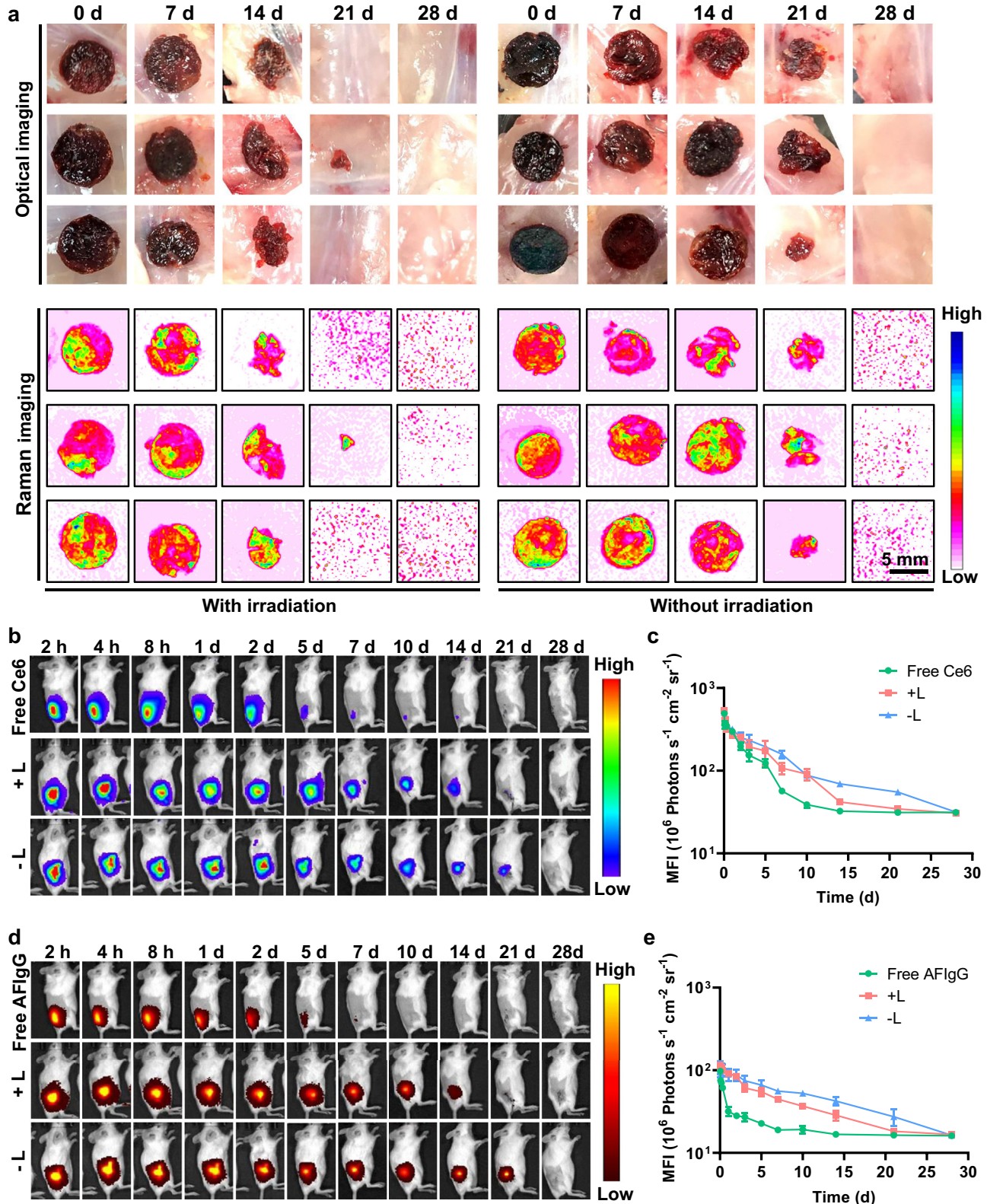

**Fig. 4 | In vivo ROS-responsive characteristics of the hydrogel. a** Optical and Raman images of Ce6@PPG samples, with and without LED irradiation, at different time points post implantation. **b** Fluorescence IVIS images depicting the in vivo retention of Ce6 fluorescence in mice with Ce6 solution (Free Ce6), AFIgG/Ce6@PPG with (+L) or without (−L) the 640 nm LED irradiation. **c** Mean fluorescence intensity (MFI) of corresponding fluorescence IVIS images in **b**.

**d** Fluorescence IVIS images depicting the in vivo retention of AFIgG fluorescence in mice injected with AFIgG solution (Free AFIgG), AFIgG/Ce6@PPG with (+L) or without (−L) the 640 nm LED irradiation. **e** Mean fluorescence intensity (MFI) of corresponding fluorescence IVIS images in **d**. LED irradiation: 640 nm, 5 mW cm$^{-2}$, 20 min on day 0, 2, 4, and 6. Data are mean ± SEM (*n* = 3 mice per group).

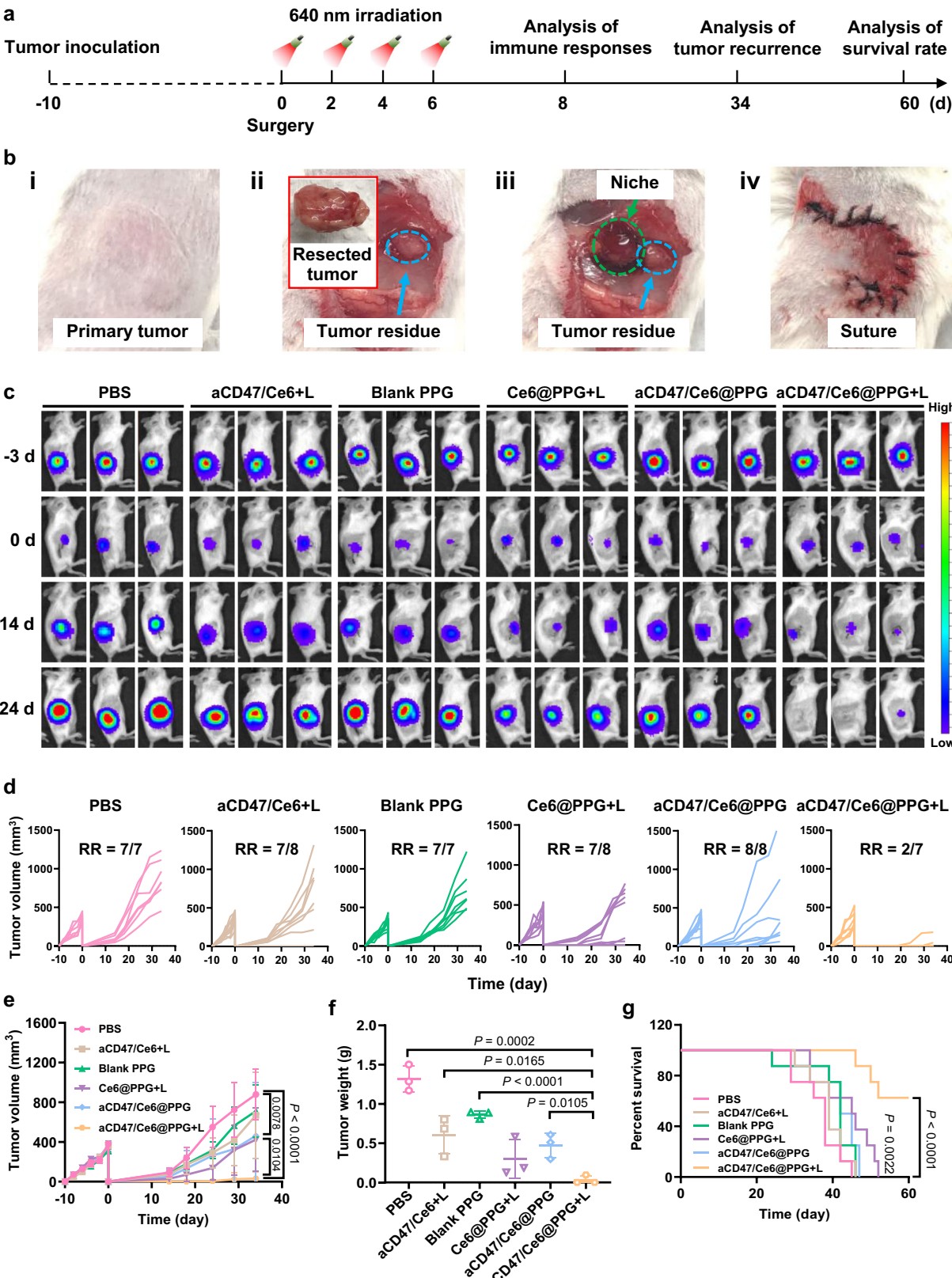

**Fig. 5 | In vivo antitumor effect on 4T1-luc tumor-bearing BALB/c mice by the aCD47/Ce6@PPG hydrogel. a** Schematic illustration of the animal experimental design. LED irradiation: 640 nm, 5 mW cm⁻², 20 min. **b** The process of the surgery in **a**, including (i) tumor volume reached ~300 mm³, (ii) tumor resection, (iii) implantation of the hydrogel (or injection of PBS or aCD47/Ce6), and (iv) suture. **c** In vivo bioluminescence imaging of tumor-resected mice receiving various treatments after surgery. Three representative mice in each treatment group were shown. Images of day 0 were taken on the day of surgery. **d** Tumor growth curves of individual mouse in different groups. RR: recurrence rate. **e** Average tumor growth curves of different groups. Data are shown as mean ± SEM (*n* = 7–8 mice per group). **f** The recurrence tumor weights of different groups on day 34 after surgery. Data are shown as mean ± SEM (*n* = 3 mice per group). **g** The survival percentages of the mice (*n* = 8 mice per group). The comparison of two groups was followed by Student's *t* test (two-tailed). Source data are provided as a Source Data file.

treated with free aCD47/Ce6, Ce6@PPG, and aCD47/Ce6@PPG were irradiated with 640 nm LED irradiation on day 0, 2, 4, and 6, respectively.

Bioluminescence signals were recorded to monitor the regrowth of tumor residues (Fig. 5c). On day 14 post the surgery, there were no significant difference in bioluminescence among the mice treated with PBS, free aCD47/Ce6 plus LED irradiation (the "aCD47/Ce6 + L" group), and blank PPG, but the mice given Ce6@PPG plus LED irradiation (the "Ce6@PPG + L" group), aCD47/Ce6@PPG, aCD47/Ce6@PPG plus LED irradiation (the "aCD47/Ce6@PPG + L" group) exhibited considerably weaker bioluminescence signals (Fig. 5c). Strikingly, the bioluminescence signals in the "aCD47/Ce6@PPG + L" group almost completely disappeared on day 24 after the surgery.

Meanwhile, we measured the direct volume change of recrudescent tumors, which was consistent with the bioluminescence analysis. The recurrence rates were larger than 87% for all the groups expect "aCD47/Ce6@PPG + L" at the end of the 34-day period (Fig. 5d). Among them, the mice in the "aCD47/Ce6 + L" and "blank PPG" groups exhibited the similar tumor regrowth to those in the "PBS" group (Fig. 5e), showing that a single administration of combined free aCD47 and Ce6-mediated PDT was ineffective. As a comparison, the tumor regrowth for the mice in the "aCD47/Ce6@PPG" group was partially suppressed, suggesting that the sustained release of aCD47 via the hydrogel was essential to maintain its antitumor effect. The inhibition on tumor recurrence was moderate for the mice in the "Ce6@PPG + L" and "aCD47/Ce6@PPG" groups, showing that the efficacy of sole PDT or aCD47 was insufficient. Markedly, the tumor recurrence in the "aCD47/Ce6@PPG + L" group was almost completely alleviated, except that 2 out of the 7 mice in this group showed barely observable tumor tissues (Fig. 5d). In addition, the mice in this group had the lowest recrudescent tumor weight (Fig. 5f and Supplementary Fig. 8) and the highest long-term survival rate (Fig. 5g), with 100% on day 40 post implantation and over 60% on day 60. Moreover, we synthesized a polyacrylic acid (PAA)-based hydrogel by crosslinking PAA and pullulan, which did not have ROS response ability and was also loaded with aCD47 and Ce6 to form the aCD47/Ce6@PAA. Although the mice treated with "aCD47/Ce6@PAA + L" showed enhanced suppression of tumor regrowth compared with free aCD47, their tumor recurrence were significantly inferior to those received "aCD47/Ce6@PPG + L" (Supplementary Fig. 9), further evidencing the exceptional tumor recurrence inhibition by the synergistic PDT and aCD47 combination enabled by our ROS-responsive hydrogel.

## Photo-irradiated aCD47/Ce6@PPG hydrogel turns "cold" tumors "hot"

The excellent mitigation of tumor recurrence on 4T1 mouse models by the photo-irradiated aCD47/Ce6@PPG hydrogel inspired us to examine its stimulant effect on the immune system of immunologically "cold" tumors. We measured the response of immune cells in lymph nodes and tumors, as well as the typical immune cell cytokines in the serums of mice on day 8 post implantation. Flow cytometry analysis (Fig. 6a, b and Supplementary Fig. 10) showed that the percentages of tumor-infiltrating CD8$^+$ T cells, a crucial indicator of antitumor immunity response activation, in the "Ce6@PPG + L" group and "aCD47/Ce6@PPG" group were 14 folds and 6 folds higher than that in the "blank PPG" group, respectively. Markedly, this value in the "aCD47/Ce6@PPG + L" group increased to 21 folds of that in the "blank PPG" group, demonstrating the exceptional immunostimulant effect by the synergistic combination of ROS and aCD47. Interestingly, this value was also higher than that in the "aCD47/Ce6 + L" group, which unambiguously evidenced the necessity of sustained drug release and prolonged aCD47 protection by the hydrogel. Consistently, CD4$^+$ T cells in tumors (Fig. 6c) and matured DC (CD11c$^+$CD80$^+$CD86$^+$) in lymph nodes (Fig. 6d) were also significantly improved in the mice receiving the "aCD47/Ce6@PPG + L" treatment

compared to all other groups. It was noted that the "aCD47/Ce6 + L" group induced immune response effectively on day 8 post the surgery (Fig. 6b, c), which was consistent with the slow tumor regrowth within this period (Fig. 5e). However, since the solution of the free drugs diffused away gradually, the drug could not maintain an adequate concentration in the resection site for too long, and the therapeutic effect diminished accordingly. As a result, the mice in this group experienced a quick tumor recurrence starting from the third week post operation.

Reeducating TAMs into M1-like phenotype was a promising strategy to reverse the immunosuppressive TME and promote antitumor immunotherapy[40–43]. As expected, the "aCD47/Ce6@PPG + L" group exhibited the strongest ability to upregulate M1-like TAMs (CD80$^+$CD11b$^+$F4/80$^+$) and downregulate M2-like TAMs (CD206$^+$CD11b$^+$F4/80$^+$) in tumor tissues (Fig. 6e–g). Accordingly, the M1/M2 ratio in this group showed an over 10-fold increase compared to that in the "PBS" group, greatly surpassing all other groups (Fig. 6h). In addition, the "aCD47/Ce6 + L" and "Ce6@PPG + L" groups also exhibited higher M1/M2 ratios than the "PBS" group, suggesting the impact of Ce6-generated ROS upon light irradiation on TAMs repolarization. These results evidenced that ROS generated by Ce6 in PDT was able to reshape the tumor immune microenvironment and potentiate the aCD47 therapy.

We next examined the levels of tumor regulatory T cells (Tregs, CD3$^+$CD4$^+$Foxp3$^+$), which could suppress the antitumor immune responses of cytotoxic T lymphocytes and induce an immunosuppressive microenvironment[44–46]. The results demonstrated that the Tregs frequency in the "aCD47/Ce6@PPG + L" group was 2.3 folds lower than in the "PBS" group, indicating an effective reduction of tumor-associated immunosuppression (Supplementary Fig. 11). In addition, the mice in the "aCD47/Ce6@PPG + L" group showed significantly increased level of typical immune related cytokines in the plasma, including interleukin 6 (IL-6, Fig. 6i), interferon γ (IFN-γ, Fig. 6j), tumor necrosis factor α (TNF-α, Fig. 6k), and interleukin 2 (IL-2, Supplementary Fig. 12), all of which contributed to the activation of systemic T-cell response. Collectively, these results demonstrated that the synergistic combination of PDT and immunotherapy, enabled by the ROS-responsive hydrogel aCD47/Ce6@PPG, could sensitize immunogically "cold" tumors for amplied aCD47 immune therapy, and stimulate strong tumor-specific immune response for effective inhibition of tumor recurrence.

## Long-term immune memory effects

Encouraged by the excellent performance of photo-irradiated aCD47/Ce6@PPG hydrogel in inhibiting tumor recurrence and stimulating antitumor immune response, we further evaluated its effect on the long-term metastasis inhibition of the mice (Fig. 7a). From photographs (Fig. 7b and Supplementary Fig. 13) and H&E staining analysis (Fig. 7c) of whole lung tissues harvested on day 60 after the surgery, metastatic nodules were observed in all groups except for the "aCD47/Ce6@PPG + L" group. The average number of lung metastatic foci in the "aCD47/Ce6@PPG + L" group was remarkably lower than in all other groups (Fig. 7e), and it was even over 15 folds lower than that in the "PBS" group, suggesting that the sustained and controlled release of aCD47 and Ce6 by ROS-responsive degradation of the hydrogel provided long-term prevention of residual 4T1 tumor cells from migration into the lungs.

The effective inhibition of lung metastasis inspired us to assess the ability of photo-irradiated aCD47/Ce6@PPG hydrogel to suppress the growth of re-challenged tumors. We inoculated 4T1 tumor cells on both completely cured mice in the "aCD47/Ce6@PPG + L" group (Fig. 7d) and untreated normal BALB/c mice. As shown in Fig. 7f, the growth of the re-inoculated tumors on the cured mice were obviously suppressed over the 16-day period, which confirmed that photo-irradiated aCD47/Ce6@PPG hydrogel played a key role in the adaptive

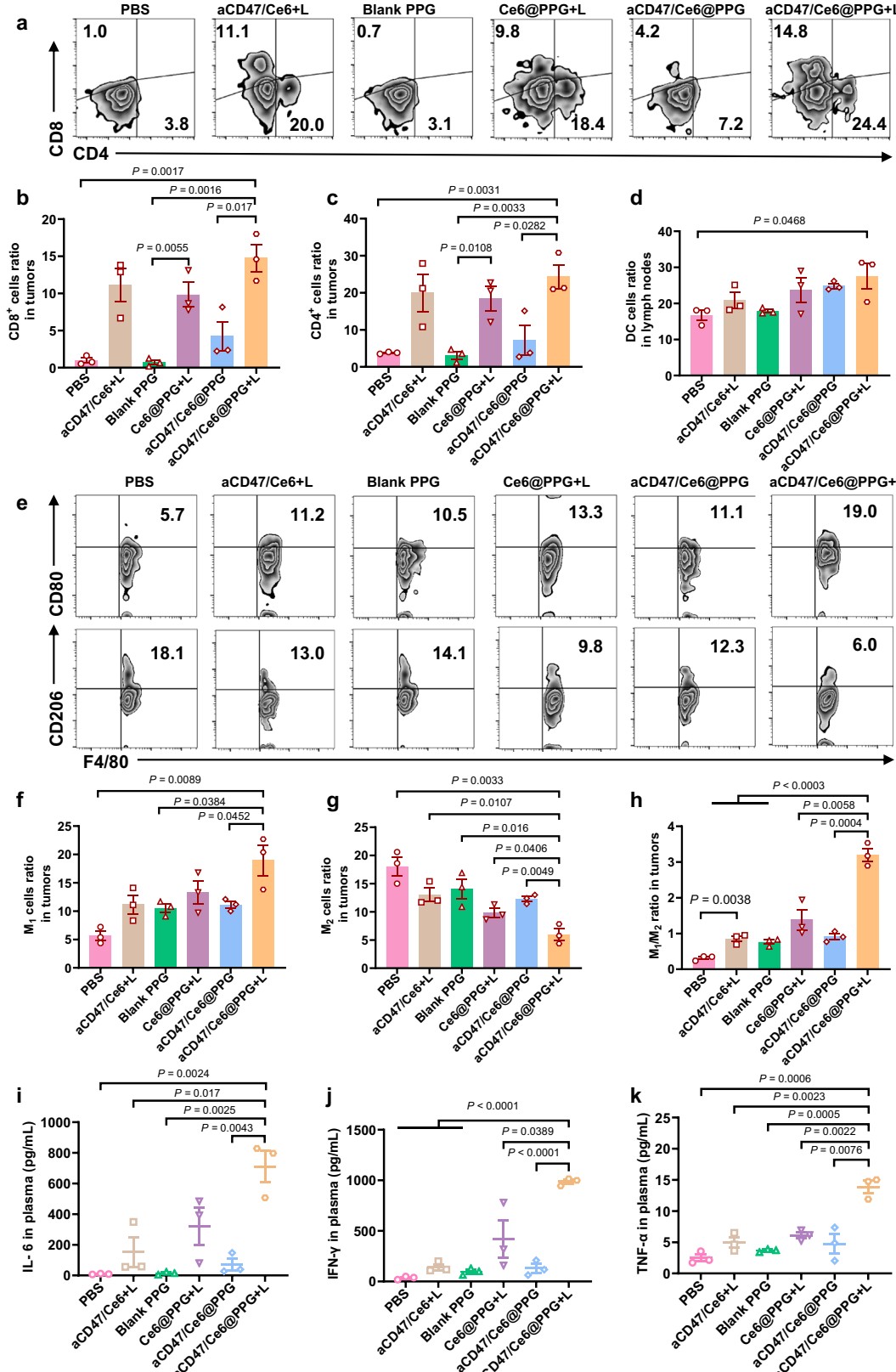

**Fig. 6 | Antitumor immune response induced by photo-irradiated aCD47/Ce6@PPG hydrogel. a** Representative flow cytometry analysis of tumor-infiltrating CD4+ and CD8+ T cells gated on CD3+ T cells on day 8 after the surgery. Relative quantifications of CD8+ T cells (**b**) and CD4+ T cells (**c**) in tumors on day 8 after the surgery. **d** Relative quantification of mature DCs in tumor-draining lymph nodes on day 8 after the surgery. **e** Representative flow cytometry analysis of M1-like TAMs (CD80+) and M2-like TAMs (CD206+) gated on F4/80+CD11b+ cells in tumors on day 8 after the surgery. Relative quantifications of the proportions of M1-like TAMs (**f**) and M2-like TAMs (**g**) gated on F4/80+CD11b+ cells. **h** The ratio of M1/M2 in different groups. Cytokine levels of IL-6 (**i**), IFN-γ (**j**), TNF-α (**k**) isolated from the plasma of the mice in different treatment groups on day 8 after the surgery. All data are shown as mean ± SEM (*n* = 3 mice per group). The comparison of two groups was followed by Student's *t* test (two-tailed). Source data are provided as a Source Data file.

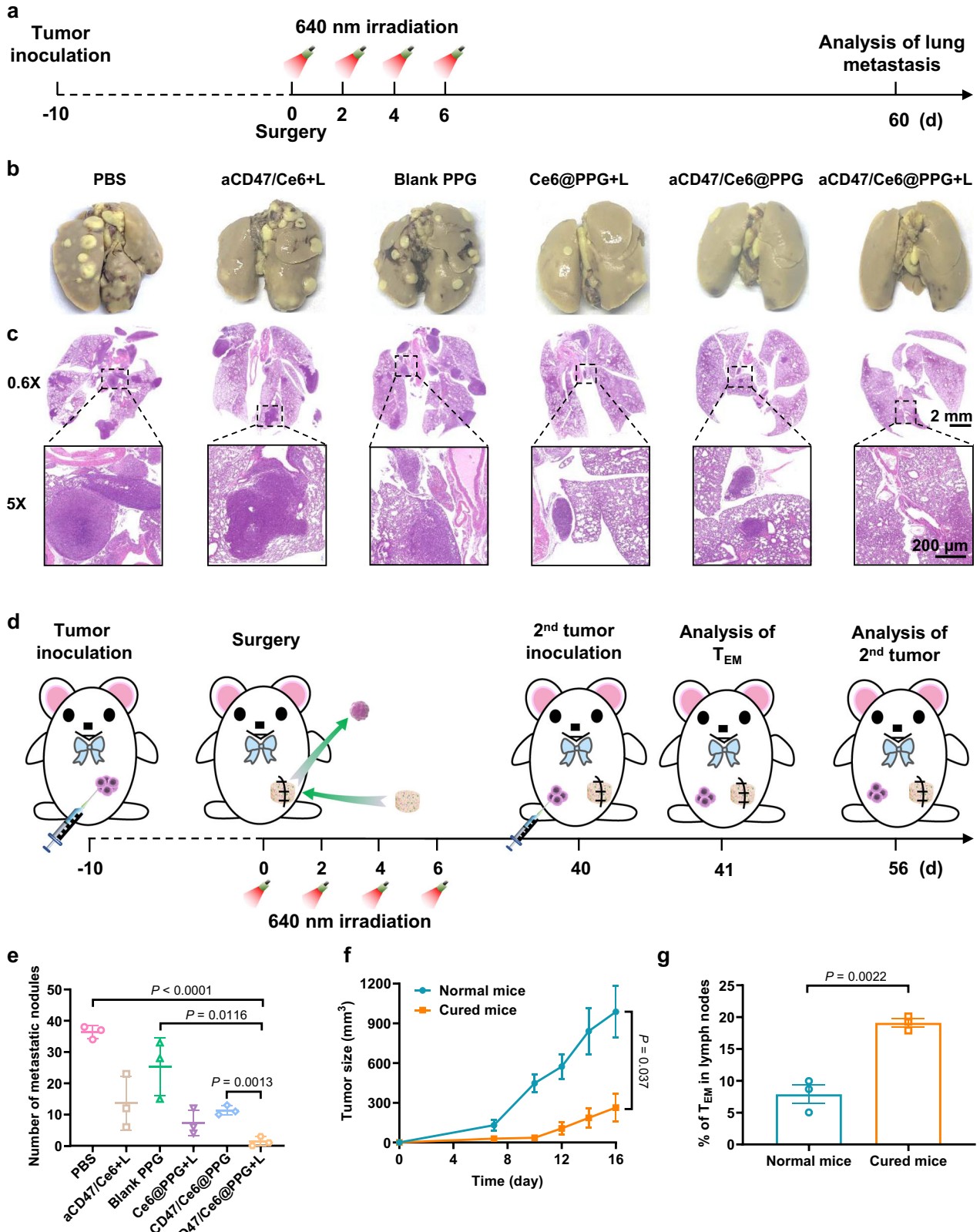

**Fig. 7 | Metastasis prevention and long-term immune effects on 4T1 tumor-bearing BALB/c mice by photo-irradiated aCD47/Ce6@PPG hydrogel.**
**a** Schematic illustration of the animal experiment design for metastasis.
**b** Representative photographs of lung tissues from mice in different groups harvested on day 60 after implantation. **c** H&E staining of lung tissues in **b**. Images are representative of six biologically independent mice. Scale bars: 2 mm (top) and 200 μm (bottom). **d** Schematic illustration of the animal experiment design on re-challenged tumors. **e** Quantification of pulmonary metastasis nodules in different groups. Data are shown as mean ± SEM ($n = 3$ mice per group). **f** Growth of the rechallenged tumors inoculated on day 40. Data are shown as mean ± SEM ($n = 5$ mice per group). **g** Relative quantification of $T_{EM}$ frequency in tumor-draining lymph nodes (gated on CD3[+]CD8[+] T cells) on day 41. Data are presented as mean ± SEM ($n = 3$ mice per group). The comparison of two groups was followed by Student's $t$ test (two-tailed). Source data are provided as a Source Data file.

systemic immunity establishment. To understand the mechanism underlying the long-term immune memory effect of the hydrogel, we further analyzed the effector memory T cells ($T_{EM}$) in lymph nodes, which could provide durable protections against tumor cell re-invasion[47,48]. As shown in Fig. 7g, the percentage of $T_{EM}$ cells in the cured mice was 2.4 folds higher than in the normal mice. In addition, immunofluorescence staining of the macrophages (F4/80) and CD8[+] T cells of the rechallenged tumors on day 16 after inoculation (Supplementary Fig. 14) also confirmed that the infiltrated populations of CD8[+] T cells and macrophages in the cured mice were significantly higher than in the normal mice, suggesting that photo-irradiated aCD47/Ce6@PPG hydrogel successfully produced tumor-specific long-term memory immunity effect that was helpful to inhibit tumor recurrence and metastasis.

## Biosafety assessment

Biocompatibility is essential for in vivo biological applications of implantable materials. As we have mentioned earlier, PPG is created by the crosslinking between a natural polysaccharide pullulan and a synthetic conjugated polymer PDDA. PDDA degrades into succinic acid, gradually yet completely, upon reaction with ROS generated either by light-irradiated Ce6 or within TME. Succinic acid is a well-known intermediate participating the tricarboxylic acid cycle in vivo[49], which dispels the concern on the biocompatibility of PDDA-based PPG. Cell culture media containing different concentrations of gel leaching solution was used to assess the cytotoxicity of PPG on NIH3T3 and 4T1 cell lines. As shown in Supplementary Fig. 15, the blank gel exhibited no cytotoxicity to both normal cells and tumor cells, indicating the good biosafety of PPG. In addition, H&E images of the skin tissues surrounding the implantation site showed no chronic inflammatory reaction after 4 weeks of treatment (Supplementary Fig. 16). Moreover, the body weight showed negligible difference among all groups (Supplementary Fig. 17) and no obvious damage and inflammation were observed on the H&E images of main organs (Supplementary Fig. 18). The complete blood panel test and serum biochemistry assays (Supplementary Fig. 19) illustrated that PPG did not cause significant change to the key hematological parameters or serum biochemical indicators after implantation. All these data confirmed that PPG possessed a good biocompatibility for in vivo applications.

## Discussion

In summary, on basis of the complete degradation of PDDA by Ce6 upon red light irradiation, we have designed and synthesized a ROS-responsive hydrogel aCD47/Ce6@PPG that co-delivers aCD47 and Ce6 for long-term synergistic combination of PDT and ICB therapy. Upon 640 nm LED irradiation, Ce6 generates ROS inside the hydrogel to trigger the gradual degradation of the hydrogel for controlled sustainable release of aCD47 and Ce6. This ROS-responsive degradation of the hydrogel, enabled by the oxidative reaction of PDDA, is essential to preserve the activity of aCD47 by annihilating the ROS that may damage aCD47. More interestingly, the hydrogel degradation as well as the consumption of the damaging ROS can be monitored by Raman imaging, given the ultrastrong and degradation-correlative Raman signal of PDDA in the cellular silent window. After being released to TME, Ce6 generates ROS upon irradiation not only for directly killing tumor cells, but also for sensitizing low-immunogenic tumors as well as potentiating the aCD47 treatment. The implanted aCD47/Ce6@PPG hydrogel completely inhibits the recurrence and metastasis of 4T1-tumor-bearing mice post operation, and the growth of re-challenged 4T1 tumors is also effectively restrained. The successful immunostimulant postsurgical treatment of this immunologically "cold" tumor implies that the ROS-responsive hydrogel holds great promise as a generally applicable localized drug-delivery platform for sustained potentiation of ICB-based cancer immunotherapy.

## Methods

### Materials

Pullulan was purchased from Meihua Group (Hebei, China). PAA (average MW of 450,000) was purchased from Aladdin Industrial Corporation Chemical (Shanghai). N-(3-Dimethylaminopropyl)-N′-Ethylcarbodiimide Hydrochloride (EDC) was purchased from Heowns Chemical (Tianjin, China). 4-Dimethylaminopyridine (DMAP) and dimethyl sulfoxide (DMSO) were purchased from Sinopharm Chemical Reagent Co., Ltd. Chlorin e6 (Ce6) was purchased from Frontier Scientific, Inc. aCD47 (Catalog no. BE0270) used in vivo was purchased from BioX Cell. AF790-labeled goat anti-mouse IgG (H&L) (Catalog no. 115-655-146) was purchased from Jackson ImmunoResear. Anti-CD3-PerCP-Cy5.5 (Catalog no. 551163), anti-CD4-APC (Catalog no. 553051), anti-CD8-FITC (Catalog no. 553030), anti-CD86-PE (Catalog no. 553692), anti-CD80-APC (Catalog no. 560016), anti-Foxp3-PE (Catalog no. 563101), anti-CD11c-FITC (Catalog no. 557400), anti-CD62L-BV421 (Catalog no. 562910), anti-CD44-APC (Catalog no. 559250) and anti-CD11b-PE-Cy7 (Catalog no. 552850) were purchased from BD Biosciences. Anti-mouse CD206-PE (Catalog no. 141705) and anti-mouse F4/80-FITC (Catalog no. 123107) were purchased from Biolegend. Human IgG ELISA Quantitation Set (Catalog no. E80-104) was purchased from Bethyl Laboratories, Inc.

### Cell lines

The metastatic murine 4T1-luc breast cancer cells and the fibroblast NIH 3T3 cells were purchased from the American Type Culture Collection. The 4T1-luc cells were maintained in Roswell Park Memorial Institute 1640 (Gibco, Invitrogen) medium with 10% fetal bovine serum (FBS), penicillin (100 U ml$^{-1}$), streptomycin (100 U ml$^{-1}$), and 1% L-glutamine. The NIH3T3 cells were cultured in complete Dulbecco's modified Eagle's medium (DMEM; Gibco, Invitrogen) with 10% FBS, penicillin (100 U ml$^{-1}$), streptomycin (100 U ml$^{-1}$), and 1% L-glutamine.

### Animals

BALB/c mice (6–8 weeks old, 18–20 g) were purchased from the Liaoning Changsheng Biotechnology Co., Ltd. (Benxi, China) and the Qinglongshan Farms (Nanjing, China). All animals were bred in a pathogen-free facility with a 12 h light/dark cycle at 20 ± 3 °C and relative humidity of 40–70%. and had ad libitum access to food and water. Animal protocols were performed under the guidelines for human and responsible use of animals in research sets approved by the Institutional Animal Care and Use Committees of Huazhong University of Science and Technology (IACUC Number: 2832) and China Pharmaceutical University (2019-08-003).

### Preparation of PPG

Poly (deca-4,6-diynedioic acid) (PDDA) was synthesized according to our previous reported procedures[28]. In total, 100 μl of PDDA solution in DMSO was added to each well of a 48-well plate. A mixed solution of pullulan (8 mg), DMAP (1 mg), and EDC (5 mg) in DMSO (200 μl) was added to the well and mixed thoroughly. After reacting for overnight, the formed PPG was taken out and dialyzed by distilled water. The PPG samples were lyophilized and stored in the dry environment. The concentration of PDDA solution was set as 20 mg ml$^{-1}$, 13.3 mg ml$^{-1}$, 10 mg ml$^{-1}$, 5.3 mg ml$^{-1}$, and 4 mg ml$^{-1}$, to prepare PPG with a PDDA/pullulan mass ratio of 1:4, 1:6, 1:8, 1:15 and 1:20, accordingly. aCD47/Ce6@PPG was prepared immediately before use. In total, 8 μl of Ce6 solution (10 mg ml$^{-1}$) and 7 μl of aCD47 solution (10 mg ml$^{-1}$) was added consequently to a lyophilized PPG to ensure each PPG was loaded with 80 μg of Ce6 and 70 μg of aCD47. The whole drug loading procedure was conducted in the dark.

### Rheological test

Rheology experiments were performed using a dynamic shear rheometer (Kinexus Rotational Rheometer, Malvern Instruments,

Malvern, UK). Under the condition of time scanning, the change of elastic modulus ($G'$) and viscous modulus ($G''$) of the gel was measured within 4 min at the frequency of 1 Hz. Under the strain scanning condition, the change of $G'$ and $G''$ in the range of 1–100% was tested at the frequency of 1 Hz.

## Swelling rate test

The mass of each lyophilized PPG sample was weighed and denoted as $m_1$. The samples were then put into pure water immediately, and took out after being soaked in water for different time (10 s, 20 s, 30 s, 45 s, 60 s, 90 s, 2 min, 3 min and 5 min). The weights of swollen PPGs were measured and denoted as $m_2$. The swelling rate was calculated as $(m_1 - m_2)/m_1 \times 100\%$.

## Scanning electron microscopy

The pore structures of PPGs with different mass ratios were characterized by SEM (SU8010, Hitachi Ltd, Tokyo, Japan). All of the samples were lyophilized, and then a broken cross-section of each sample was sputter-coated with platinum and examined at 15 kV.

## Preparation of polyacrylic acid (PAA)-based hydrogel

PAA-based hydrogel was prepared following a procedure similar to the preparation of PPG. In total, 100 μl of PAA solution (10 mg ml⁻¹) in DMSO was added to each well of a 48-well plate. A mixed solution of pullulan (8 mg), DMAP (1 mg), and EDC (5 mg) in DMSO (200 μl) was added to the well and mixed thoroughly. After reacting for overnight, the formed gel was taken out and dialyzed by distilled water. The concentration of PAA solution was fixed at 10 mg ml⁻¹ to form a hydrogel with a PAA/pullulan mass ratio of 1:8.

## Degradation of PDDA hydrogel at varied ROS levels

We used the reaction between NaClO and $H_2O_2$ to approach a finely controlled singlet oxygen level. A PPG hydrogel (PDDA/pullulan mass ratio 1:8) was placed in each well (1–6) containing 1% $H_2O_2$. In total, 0, 1, 2, 4, 10, 20 μl of NaClO (5%) were added to well 1, 2, 3, 4, 5, and 6 each day, respectively. The photograph showing the status of PPG was taken at different time after the experiment started.

## In vivo degradation behavior

To evaluate the in vivo degradation rates of PPGs with different mass ratios, the PPG sample was implanted into the surgical bed of each tumor-resected mouse. On day 10 and day 20, one of the mice in each group was sacrificed and the implanted PPG was dissected for analysis.

To test the ROS-responsive degradation of PPG, 30 tumor-resected mice were divided into two groups ("with light" group and "without light" group), and Ce6@PPG containing 80 μg of Ce6 was implanted into the surgical bed of each mouse. For the "with light" group, each mouse was irradiated by a 640 nm LED light at the surgical site (5.0 mW cm⁻², 20 min) at 4 h, 2, 4 and 6 days post implantation, whereas the "without light" group received no irradiation. For both groups, three randomly selected mice were sacrificed on day 0, 7, 14, 21, and 28, respectively, and the implantation site were dissected for analysis by optical and Raman imaging.

## Raman imaging

Raman imaging was conducted with a Bruker SENTERRA II confocal Raman microscope, using a 100 μm step size and 150 × 150 points using fast mapping mode. In total, 785 nm laser beam with 10 mW power was used for excitation. Signal acquisition time of each point was fixed at 100 ms and the spectra was collected from 400 to 3000 cm⁻¹. The biorthogonal Raman signal of C≡C at 2120 cm⁻¹ was used for imaging presenting.

## In vitro drug release study

The IgG/Ce6@PPG containing 80 μg of Ce6 and 100 μg of IgG was placed in each of the 6 vials. The samples were divided into two groups ("with light" group and "without light" group), and 3.0 ml of PBS (pH 7.4) was added in each vial as the drug release media. The samples in the "with light" group was irradiated with a 640 nm LED light (5.0 mW cm⁻²) for 20 min at 4 h, 2, 4 and 6 days after sampling. At 1 h, 2 h, 4 h, 8 h, 1 day, 2 days, 3 days, 5 days, 7 days, 10 days and 15 days, 1 ml of the media in each vial was removed for further analysis and another 1 ml of fresh media was added afterwards. The released amount of IgG was measured using Enzyme linked immunosorbent assay (ELISA).

## In vivo drug release study

To evaluate the in vivo drug release of Alexa Fluor 790 AffiniPure Goat Anti-Mouse IgG (AF-IgG) and Ce6, the AF-IgG/Ce6@PPG hydrogel was implanted into the surgical bed of each tumor-resected mouse. The solution of AF-IgG or Ce6 was administered into the incision sites as a control. Each mouse in irradiation-receiving groups was then irradiated with a 640 nm LED light (5.0 mW cm⁻²) for 20 min at 4 h, 2, 4 and 6 days. Fluorescence images were taken at the desired interval and analyzed by a Living Imaging software.

## In vitro study of protein damage by ROS

BSA (66 kDa, purchased from Solarbio) was used as a model protein. In total, 200 μg BSA was dissolved in 1 ml DI water and 10 μg Ce6 in DMSO 1 μl was added to the BSA solution under vigorous stirring to form a homogenous dispersion. The dispersion was irradiated (640 nm, 5 mW cm⁻²) for 5 min or 60 min to reach partial or complete oxidative denaturation. For the evaluation of protein protection effect of PDDA, 1 mg PDDA was added to the mixture before light irradiation. For the group with 60 min light irradiation, 6 mg PDDA was added to the mixture in six portions with 10 min interval. The mixture after light irradiation was diluted with 10-fold DI water and then filtered to remove insoluble substances and analyzed with SDS-PAGE. Separation gel (10%) was prepared with the following recipe: $H_2O$ 4.1 ml, acrylamide/bis (30%) 3.3 ml, Tris-HCl (1.5 M, pH 8.8) 2.5 ml, SDS, 10% 100 μl, N,N,N′,N′-tetramethylethylene-diamine (TEMED) 10 μl and ammonium persulfate (APS), 10% 32 μl. The stacking gel (4%) contains $H_2O$ (6.1 ml), acrylamide/bis (30%, 1.3 ml), Tris-HCl (1.0 M, pH 6.8, 2.5 ml), SDS (10% 100 μl), N,N,N′,N′-tetramethylethylene-diamine (10 μl), and APS (10% 100 μl). Individual protein sample (20 μl) was mixed with the loading buffer (5 μl) and incubated at 100 °C for 10 min, before being loaded for separation by electrophoresis.

## NanoSPR

A mixed solution of aCD47 (500 μg ml⁻¹) and Ce6 (40 μg ml⁻¹) was divided into two identical aliquots. One was used as prepared and the other was added with an aqueous solution of PDDA (70 μg ml⁻¹). Each aliquot was placed in a 2 ml tube and received continuous LED light irradiation (640 nm, 5 mW cm⁻²) for 2 h. The 96-well microarray chip plate of NanoSPR modified with CD47 antigen was used for affinity detection. PBS was used as control treated in the same conditions.

## In vivo tumor models and treatment

The antitumor study was performed on 4T1 tumor-bearing mouse models. In order to mimic the situation of incomplete tumor removal in clinic, 4T1-luc breast cancer cells (1 × 10⁶) were inoculated into the flanks of mice, and -90% of the tumor was surgically resected when the tumor volume reached -300 mm³. The mice were randomly grouped and each was treated with PBS solution, a mixed solution of aCD47 and Ce6 (free aCD47/Ce6), blank PPG, Ce6@PPG, or aCD47/Ce6@PPG on the resection site before suture (Fig. 4b). The dose administered on each mouse was set as 80 μg for Ce6 and 70 μg for aCD47. The surgical

beds on the mice treated with free aCD47/Ce6, Ce6@PPG, and aCD47/Ce6@PPG were irradiated with 640 nm LED on day 0, 2, 4, and 6, respectively. The mice were monitored by an IVIS and the tumor volume was calculated with the following equation: tumor volume = width$^2$ × length × 0.5. On day 34 post implantation, three mice in each group were sacrificed, and the tumor residues were dissected for analysis. The survival time and body weights of the remaining mice were continued to be recorded until day 60 post tumor resection. Finally, all the major organs, including heart, liver, lung, spleen, and kidney, were collected and examined bouty H&E staining.

### In vivo tumor treatment with free aCD47 and aCD47/Ce6@PAA hydrogel

The antitumor study was performed on 4T1 tumor-bearing mouse models. To mimic the situation of incomplete tumor removal, 4T1 breast cancer cells (1 × 10$^6$) were inoculated into the flanks of mice, and ~90% of the tumor was surgically resected when the tumor volume reached ~300 mm$^3$. The mice were randomly grouped and each was treated with PBS solution, a solution of aCD47, aCD47/Ce6@PAA or aCD47/Ce6@PPG on the resection site before suture. The dose administered on each mouse was set as 80 μg for Ce6 and 70 μg for aCD47. aCD47 was applied without dilution to lyophilized PPG to form aCD47/Ce6@PAA and aCD47/Ce6@PPG. For aCD47 solution, aCD47 was diluted six times before injection. The surgical beds on the mice treated with aCD47/Ce6@PAA and aCD47/Ce6@PPG were irradiated with 640 nm LED (5 mW cm$^{-2}$, 20 min) on day 0, 2, 4, and 6, respectively. The tumor volume was traced and calculated as: width$^2$ × length × 0.5. On day 34 post implantation, all mice in each group were sacrificed, and the tumor residues were dissected.

### Flow cytometry analysis of immune cells

On day 8 post surgery, another three mice from each group mentioned above were sacrificed, and the tumor residues and lymph nodes were surgically resected. Tumor-infiltrating lymphocytes (TILs) were collected and incubated with anti-CD3-PerCP-Cy5.5 (1/50 dilution), anti-CD4-APC (1/200 dilution), and anti-CD8-FITC (1/200 dilution) antibodies according to the standard protocols, and the contents of CD4$^+$ and CD8$^+$ T cells in the tumors were analyzed by a flow cytometer (BD FASCVerse, USA). The regulatory T cells (Tregs) were examined by staining the TILs with anti-CD3-PerCP-Cy5.5 (1/200 dilution), anti-CD4-APC (1/200 dilution), and anti-Foxp3-PE (1/200 dilution) antibodies, and analyzed by the flow cytometer. The macrophage was examined by staining the TILs with anti-CD11b-PE-Cy7 (1/200 dilution), anti-F4/80-FITC (1/200 dilution), anti-CD206-PE (1/200 dilution), and anti-CD80-APC antibodies (1/200 dilution), and analyzed by the flow cytometer. The lymph nodes nearby the tumors were collected at the same time to study the matured DCs in tumor-draining lymph nodes. After immunofluorescence staining with anti-CD11c-FITC (1/200 dilution), anti-CD80-APC (1/200 dilution), and anti-CD86-PE antibodies (1/200 dilution), the samples were analyzed by the flow cytometer. All the data were analyzed using the FlowJo software.

### Cytokine analysis

Serum samples were isolated from the mice sacrificed on day 8 and diluted for analysis. Tumor necrosis factor (TNF-α), interferon gamma (IFN-γ), interleukin 6 (IL-6), and interleukin 6 (IL-2) were analyzed by ELISA kits according to vendors' instructions (KeyGEN Biotech, China).

### Lung metastasis analysis

On day 60 post surgery, three remaining mice from each group were sacrificed and their lung tissues were harvested and fixed in Bouin's solution for metastasis analysis. The yellow nodules on the surface of lungs indicating the tumor metastasis sites were counted visually and examined by H&E staining.

### Re-challenged tumor models and evaluation of long-term immune memory

On day 40 post the surgery, six completely cured mice in the "aCD47/Ce6@PPG + L" group and six healthy mice were inoculated with 4T1 cells (5 × 10$^5$) on the left flanks. On day 41, three mice in each group were sacrificed and the lymphocytes were extracted from their lymph nodes. The effector memory T cells (T$_{EM}$ cells) were examined by staining the lymphocytes with anti-CD3-PerCP-Cy5.5, anti-CD8-FITC, anti-CD62L-BV421, and anti-CD44-APC antibodies and analyzed by flow cytometry. The tumor volumes of the rest mice were measured in the following 16 days.

### Cytotoxicity evaluation in vitro

PPGs were incubated in DMEM for 24 h. The NIH3T3 and 4T1 cells were seeded in 96-well plates and incubated in DMEM for overnight. The leachates of PPGs in DMEM were diluted sequentially (100, 50, 25, 12.5 and 6.25%) and incubated with both NIH3T3 and 4T1 cells for different time (24 and 48 h). The cell viability was then evaluated by MTT assays.

### Blood panel test and serum biochemistry assay

The aCD47/Ce6@PPG niche was implanted into individual healthy mouse, and each mouse was then irradiated with a 640 nm LED light (5.0 mW cm$^{-2}$) for 20 min at 4 h, 2, 4 and 6 days. The whole blood and serum of each mouse were then collected on day 0, 10, and 31 respectively for the blood panel test and serum biochemistry assay. The levels of ALT, ALP, HGB, RBC and WBC were measured by standard clinical methods.

### Statistical analysis

All the analysis data are given as mean ± SEM. The results were analyzed by the Student's $t$ test between two groups. Exact $p$ values were provided accordingly in the figures.

### Reporting summary

Further information on research design is available in the Nature Research Reporting Summary linked to this article.

## Data availability

All data generated or analyzed during this study are included in this published article and its Supplementary Information file and the Source Data file. Source data are provided with this paper.

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

## Acknowledgements

We thank Dr. Shun Wang for help on Raman spectroscopy, and Dr. Chenghui Wang for help on cellular measurement. This work was funded by the National Natural Science Foundation of China (21877042, 22077038, 22107032); the National Basic Research Plan of China (2018YFA0208903); Postdoctoral Research Foundation of China (2017M622454 and 2020T130038ZX); and Huazhong University Startup Fund.

## Author contributions

These authors contributed equally: Y.Z., S.T., L.H. and Y.Li. Y.Z. and S.T. designed the synthesis methods and prepared samples. Y.Z., S.T. and L.H. performed the in vitro sample characterization. Y.Z., L.H., Y.Li, and Y.Lu designed and performed the in vivo experiments. Y.Lu and H.L. participated in the synthesis. G.C. assisted with the Raman imaging. G.L.L. assisted with the SPR experiments. F.M., X.Y., J.T., C.S. and L.L. designed and supervised the experiments. S.T. and L.L. conceived and obtained funding for the project. L.L. oversaw the research. Y.Z., S.T., L.H., and L.L. wrote the paper. All authors discussed the results and have given approval to the final version of the paper.

## Competing interests

The authors declare no competing interests.
