## [Peer Review File · Nature Communications]

REVIEWER COMMENTS

Reviewer #1 (Remarks to the Author):

In this manuscript, Lou and coworkers have used a hydrogel based on poly(deca-4,6-diynedioic acid)(PDDA) for delivering Ce6 and CD47 antibody for synergistic cancer therapy. The authors emphasized that the hydrogel could absorb the ROS generated by exposing Ce6 to light and oxidize and degrade to release the encapsulated drug. Also, the authors emphasized that the hydrogel could protect the antibody from oxidation by ROS, therefore maintaining the antibody's activity. In addition, the hydrogel could be monitored via Raman imaging. Even though the overall design has made promising therapeutic efficacy, the authors have reported PDDA for Raman probes (Nat. Commun. 2020, 11, Article number: 81). Also, ROS responsive hydrogels have already been reported for controlled drug delivery. Combining photodynamic therapy with ICB has already been utilized to enhance cancer immune therapy. Thus, this work may not be suitable for publication on Nature Communication. A few major issues need to be addressed before publication elsewhere.

Major issues:

1. The degradation of PDDA hydrogel at varied ROS levels needs to be evaluated.
2. A control hydrogel encapsulated Ce6 and aCD47 needs to be studied. This hydrogel should not have ROS responsive ability.
3. Evidence demonstrating that the PDDA hydrogel could protect the antibody needs to be provided.

Reviewer #2 (Remarks to the Author):

In the manuscript, the authors created a new ROS-responsive hydrogel to co-deliver ICB antibodies and photosensitizers. The hydrogel skeleton PDDA can protect ICB antibodies by scavenging the harmful ROS, and synchronously trigger the gradual degradation of the hydrogel to release the drugs in a controlled manner. The system obtained good tumor inhibition in mouse models. This work represents a new ROS-responsive degradation mechanism that may inspire future drug delivery research. It is also interesting to know that the degradation process can be monitored by Raman imaging, attributed to the ultrastrong and degradation-sensitive Raman signaling of PDDA, which is unprecedented. Given the novelty and comprehensive characterization work in this study, I would recommend accepting this manuscript in Nature Communications. The manuscript is well written, but some minor issues need to be addressed before publication.

1. The complete degradation of PDDA in the presence of light-irradiated Ce6 is interesting. Is the mechanism the same as the degradation in sunlight, as what they have reported before? Are there any other byproducts?
2. From Suppl. Fig. 3, in addition to ROS-responsive degradation, it seems that the in vivo degradation of pullulan also affects the hydrogel degradation. The authors need to discuss the influence of pullulan.
3. The hydrogel PPG was used for postsurgical prevention of tumor recurrence. Is it possible that this material be used for direct tumor inhibition?
4. The authors need to identify whether there is a photothermal effect of Ce6 in the strategy, in that PTT may also activate the immune system.
5. There are some typos in the manuscript. Please check through the manuscript to ensure they have been corrected.

Reviewer #3 (Remarks to the Author):

In this very interesting study by Zhang et al, the authors described a strategy of treating cancer, by combining immune checkpoint inhibitors and photodynamic therapy. They generated a ROS-

responsive hydrogel PPG by crosslinking PDDA and pullulan, to realize sustained delivery of CD47-blocking antibodies and PS. ROS generated during PDT which could lead to degradation of antibodies reacted with PDDA and thus improved the efficacy of antibodies. This study focused on a very important topic to address critical questions in cancer immunotherapy. The manuscript was well-written and the data were well organized and presented. Despite the enthusiasm, several concerns need to be addressed to further improve the study.

1. There seems to be a concern regarding the dose of the functional antibodies can be released from the hydrogel. According to supplementary Fig4, only less than 15% of loaded antibodies can be released in the in vitro assay, with or without LED irradiation. In addition, can the authors quantify the dose of loaded antibody that can be released in the in vivo setting?
2. Would co-incubation with Ce6 induce degradation of CD47 antibody?
3. In the in vivo experiments shown in Fig4 and Fig5, the aCD47/Ce6 showed a strong effect on increasing CD8 and CD4 T cell infiltration but demonstrated no effects on inhibiting tumor growth. The authors need to comment on this.
4. In Fig5, M1 vs M2 macrophages were defined only based on CD80 and CD206 expression. Additional markers should be included to more precisely define M1 and M2 macrophages.
5. In Fig5, the authors showed exciting efficacy of the aCD47/Ce6@PPG+L combination therapy. However, a direct comparison between the combination therapy and aCD47 alone is missing. In the two groups included in the experiments (aCD47/Ce6+L and aCD47/Ce6@PPG), it's possible that the efficacy of CD47 Ab was compromised due to ROS generated by Ce6. The author should directly compare the combination therapy (aCD47/Ce6@PPG+L) and aCD47 alone to demonstrate the efficacy of combination therapy is superior.
6. More detailed description regarding the lung metastasis model should be provided. Mice numbers used for the in vivo experiments should be provided.
7. Experimental procedure of loading aCD47 and Ce6 to PPG should be included in the methods section with more details.

Reviewer #1 (Remarks to the Author):

In this manuscript, Lou and coworkers have used a hydrogel based on poly(deca-4,6-diyndioic acid)(PDDA) for delivering Ce6 and CD47 antibody for synergistic cancer therapy. The authors emphasized that the hydrogel could absorb the ROS generated by exposing Ce6 to light and oxidize and degrade to release the encapsulated drug. Also, the authors emphasized that the hydrogel could protect the antibody from oxidization by ROS, therefore maintaining the antibody's activity. In addition, the hydrogel could be monitored via Raman imaging. Even though the overall design has made promising therapeutic efficacy, the authors have reported PDDA for Raman probes (Nat. Commun. 2020, 11, Article number: 81). Also, ROS responsive hydrogels have already been reported for controlled drug delivery. Combining photodynamic therapy with ICB has already been utilized to enhance cancer immune therapy. Thus, this work may not be suitable for publication on Nature Communication. A few major issues need to be addressed before publication elsewhere.

Response: We really appreciate the reviewer for the elaborate review and helpful comments. We also thank the reviewer for acknowledging that the overall design in this study has made promising therapeutic efficacy. In addition, we agree with the reviewer that PDDA for Raman probes, ROS responsive hydrogels for controlled drug delivery, and combining PDT with ICB have been reported previously. However, in the manuscript we are submitting here, we report *the creation of an entirely new ROS-responsive hydrogel PPG*, employing PPDA as the scaffold skeleton, to realize sustained co-delivery of Ce6 and aCD47. There have been many deliberately designed ROS-responsive hydrogels, but PPG undergoes *an unprecedented ROS-scavenging mechanism and yields biocompatible succinic acid after degradation*. More strikingly, PDDA has an ultrastrong and degradation-correlative Raman signal of PDDA in the cellular silent window. We did report PDDA for Raman probes earlier [Nature Communications, 2020 (11), 81], but in this manuscript, we emphasize the further utilization of this intriguing feature of PDDA to *monitor the ROS-responsive hydrogel degradation and the consumption of the damaging ROS in real time by Raman imaging*. To the best of our knowledge, this is the first time that Raman imaging is used to trace the activity and faith of drug carriers in vivo.

On the other hand, as the reviewer has pointed out, combining with PDT has been reported as a promising strategy to potentiate ICB therapy. ROS generated in PDT can reverse the immunosuppressive tumor microenvironment and sensitize “immune-cold” tumors for enhanced ICB therapy. However, excessive ROS, especially ROS generated simultaneously by photosensitizers (PSs) in confined carriers, can inevitably destruct co-delivered antibodies. We have demonstrated in our manuscript that the activity of aCD47 was significantly restrained in the presence of light-irradiated PSs. The PPG we report here represents a conceptually new drug delivery system that can *preserve the activity of aCD47 by scavenging the damaging ROS generated during PDT*, and more favorably, *utilize the ROS consumption to realize the controlled release of aCD47 and Ce6 for long term synergy of cancer immunotherapy and PDT*.

Collectively, although the Raman features of PDDA and ROS-responsive hydrogels

have been reported before, the creation of PDDA-based ROS-scavenging hydrogels for sustained potentiation of ICB by PDT is completely innovative. Using Raman imaging to monitor hydrogel degradation and ROS scavenging is also unprecedented. In addition, the detailed investigation and solid evidence on the promising therapeutic efficacy of our PPG should inspire widely exploiting effective strategies to realize long term potentiation of ICB therapies.

We apologize for any unclearness the previous version of manuscript might cause, and we believe our explanation could better illustrate the novelty and highlights of this work. In addition, we also greatly appreciate the valuable comments raised by the reviewer, which we do believe will improve the quality of our manuscript. We have herein carried out a series of additional experiments to fully address these concerns and critiques.

Major issues

1. The degradation of PDDA hydrogel at varied ROS levels needs to be evaluated.

Response: We really thank the reviewer for this comment. As the reviewer suggested, we have conducted additional experiments to investigate the degradation behavior of PPG at varied singlet oxygen levels. We used the reaction between NaClO and H₂O₂ to approach a finely controlled singlet oxygen level. As shown in Fig. R1, a PPG hydrogel was placed in each well (1–6) containing 1% H₂O₂, and different amount of NaClO was added into the corresponding well at various amounts daily. As the concentration of NaClO increased, the time required for complete degradation of the gel gradually decreased, demonstrating the accelerated degradation of the PPG hydrogels in the presence of increased ROS levels. This study has been added in the revised manuscript (Page 9, highlighted in red) and the Supplementary Information (Methods and Supplementary Fig. 3, highlighted in red).

Fig. R1 Degradation procedure of PPG at various ROS levels. A PPG hydrogel (PDDA/pullulan mass ratio 1:8) was placed in each well containing 1% H₂O₂ solution, which was then added with different amounts of NaClO on a daily basis. The photograph showing the status of PPG was taken at different time after the experiment started. For these wells, 0, 1, 2, 4, 10, 20 μL of NaClO (5%) were added to Well **1, 2, 3, 4, 5, and 6** every day, respectively.

2. A control hydrogel encapsulated Ce6 and aCD47 needs to be studied. This hydrogel should not have ROS responsive ability.

Response: Thank the reviewer for this very helpful suggestion. This is a very good point to validate the role of the ROS responsive ability of PPG hydrogels. To prepare such a control hydrogel, we synthesized a polyacrylic acid (PAA)-based hydrogel by crosslinking PAA and pullulan, following the same protocol of preparing the PPG hydrogel. The PAA hydrogel did not have the ROS responsive ability, and was also loaded with aCD47 and Ce6 to form the aCD47/Ce6@PAA. To evaluate the therapeutic effects of aCD47/Ce6@PAA and aCD47/Ce6@PPG, we used the incompletely resected 4T1-tumor mouse models and investigated the suppression of tumor recurrence after surgery (Fig. R2). PBS was used as a blank control and all mice received irradiation (640 nm, 5 mW cm⁻², 20 min) on the dissection site on Day 0, 2, 4, and 6 after the surgery. The tumor regrowth for the mice treated with “aCD47/Ce6@PAA+L” showed partially inhibition. As a comparison, the tumor recurrence in the “aCD47/Ce6@PPG+L” group was almost completely alleviated, and only 3 out of the 8 mice in this group showed barely observable tumor tissues, clearly evidencing the superior anti-tumor effect of the ROS-responsive aCD47/Ce6@PPG. We have added this study to the revised manuscript (Page 18–19, highlighted in red) and the Supplementary Information (Supplementary Methods and Supplementary Fig. 9, highlighted in red).

Fig. R2 In vivo antitumor effect on 4T1-luc tumor-bearing BALB/c mice by PBS, aCD47/Ce6@PAA+L, and aCD47/Ce6@PPG+L. **a** Average tumor growth curves of mice receiving different treatments, Data are shown as mean ± SEM (n = 8). The comparison of two groups was followed by student’s t-test (two-tailed). * *P* < 0.05, ** *P* < 0.01, and *** *P* < 0.001. **b** The photograph of recurrent tumors collected from the mice receiving different treatments on Day 34 post surgery.

3. Evidence demonstrating that the PDDA hydrogel could protect the antibody needs to be provided.

Response: Thanks so much for the valuable suggestion. One of the most important features of PPG is its protection on the antibody. We have demonstrated in the manuscript that PDDA was effective in protecting protein BSA from ROS-triggered damage by gel electrophoresis (Supplementary Fig. 6 in the revised Supplementary Information). PDDA could also preserve the activity of aCD47 by consuming the ROS generated by Ce6, as evidenced by the Nano Surface Plasmon Resonance (NanoSPR) experiment (Supplementary Fig. 7 in the revised Supplementary Information). To further examine the antibody protective effect of PPG hydrogel, as the reviewer suggested, we used NanoSPR to investigate the change of binding affinity of aCD47 antibody and CD47 antigen in the presence and absence of PPG. As shown in Table R1, without PPG hydrogel, the antibody-antigen dissociation rate (K_d) of free aCD47 greatly increased to $3.04 \times 10^{-4} \text{ s}^{-1}$ and the antibody-antigen effective binding concentration K_D increased to $4.59 \times 10^{-9} \text{ M}$ when Ce6 and light irradiation were applied, demonstrating the deactivation effect of the Ce6-generated ROS on aCD47. However, in the presence of PPG, the value of K_d decreased to $5.79 \times 10^{-5} \text{ s}^{-1}$ and that of K_D decreased to $2.51 \times 10^{-9} \text{ M}$, indicating that the PPG hydrogel could effectively preserve the activity of aCD47 from damaging ROS. We have added this study to the revised manuscript on Page 16 (highlighted in red) and the Supplementary Information (Supplementary Table 1, highlighted in red).

Samples	$K_a (\text{M}^{-1}\text{s}^{-1})$	$K_d (\text{s}^{-1})$	$K_D (\text{M})$
aCD47+L	8.32×10^4	8.47×10^{-5}	1.02×10^{-9}
aCD47+Ce6	4.87×10^4	5.30×10^{-5}	1.09×10^{-9}
aCD47+Ce6+L	6.61×10^4	3.04×10^{-4}	4.59×10^{-9}
aCD47+Ce6+PPG+L	2.31×10^4	5.79×10^{-5}	2.51×10^{-9}

Table R1 The binding affinity of aCD47 antibody and CD47 antigen measured using the NanoSPR technology under different conditions.

Reviewer #2 (Remarks to the Author):

In the manuscript, the authors created a new ROS-responsive hydrogel to co-deliver ICB antibodies and photosensitizers. The hydrogel skeleton PDDA can protect ICB antibodies by scavenging the harmful ROS, and synchronously trigger the gradual degradation of the hydrogel to release the drugs in a controlled manner. The system obtained good tumor inhibition in mouse models. This work

represents a new ROS-responsive degradation mechanism that may inspire future drug delivery research. It is also interesting to know that the degradation process can be monitored by Raman imaging, attributed to the ultrastrong and degradation-sensitive Raman signaling of PDDA, which is unprecedented. Given the novelty and comprehensive characterization work in this study, I would recommend accepting this manuscript in Nature Communications. The manuscript is well written, but some minor issues need to be addressed before publication.

1. The complete degradation of PDDA in the presence of light-irradiated Ce6 is interesting. Is the mechanism the same as the degradation in sunlight, as what they have reported before? Are there any other byproducts?

Response: We are really thankful that the reviewer has a positive opinion about our work. We also appreciate the valuable comments the reviewer raised. According to Fig. R3 (Fig. 1 in the revised manuscript) and our previous work¹, the mechanism of the degradation of PDDA in the presence of light-irradiated Ce6 is the same as that of the PDDA degradation in sunlight. Upon sunlight irradiation, PDDA generates ROS to trigger oxidative degradation, yielding the major degradant of succinic acid and other small molecules. In the presence of light-irradiated Ce6, Ce6 generates ROS to oxidize C=C and C≡C bonds in PDDA backbones, leading to the complete degradation of PDDA polymer (Fig. R3a, R3b, R3c, R3d). From ¹H NMR (Fig. R3e), ¹³C NMR (Fig. R3f), and HR-MS (Fig. R3g), the dominant degradation product of PDDA was still succinic acid. There are some other byproducts from the PDDA degradation, but the amount of individual byproduct is very limited and invisible on these spectra.

Fig. R3 Complete degradation of PDDA by Ce6 upon red light irradiation. **a** Degradation reaction of PDDA in the presence of Ce6 with the irradiation. **b** Absorption spectra of the mixed solution of PDDA and Ce6 after exposure to the irradiation for different time. **c** Raman spectra of the mixed solution of PDDA and Ce6 after exposure to the irradiation for different time. **d** Changes in the relative intensities of PDDA absorbance at 460 nm ($A_{460 \text{ nm}}$, Raman at 1522 cm^{-1} (Raman $I_{1522 \text{ cm}^{-1}}$) and 2121 cm^{-1} (Raman $I_{2121 \text{ cm}^{-1}}$) as a function of irradiation time. **e**, **f**, **g** ^1H NMR (**e**), ^{13}C NMR spectra (**f**), and HR-MS spectra (**g**) of PDDA degradants, in comparison with a succinic acid standard. The red light irradiation: 640 nm , 5 mW cm^{-2} .

2. From Suppl. Fig. 3, in addition to ROS-responsive degradation, it seems that the *in vivo* degradation of pullulan also affects the hydrogel degradation. The authors need to discuss the influence of pullulan.

Response: We really appreciate the reviewer for this valuable critique and for the helpful suggestions. As shown on Page 9–10 in the revised manuscript, the *in vivo* degradation of pullulan is also a major factor of the hydrogel degradation. As a natural polysaccharide with good biocompatibility and biodegradability²⁻³, pullulan can also degrade in physiological conditions. This is also one of the reasons why we choose pullulan to construct the hydrogel. As shown in Fig. R4a (Supplementary Fig. 4 in the revised Supplementary Information), the blank PPGs with PDDA/pullulan mass ratios

of 1:15 and 1:20 could still complete the degradation in around 3 weeks without loading Ce6. For drug-loaded PPGs at a PDDA/pullulan mass ratio of 1:8, they completely degraded on Day 28 if no irradiation was applied (Fig. R4b, Fig 3a in the revised manuscript). These processes clearly demonstrated the metabolic degradation of pullulan in vivo. As expected, when irradiation was applied, the degradation of Ce6@PPG was significantly accelerated (Fig. R4c, Fig 3a in the revised manuscript), which evidenced that both oxidative degradation of PDDA and metabolic degradation of pullulan contributed to the degradation of PPGs. We have added this discussion in the revised manuscript (Page 9–10, highlighted in red).

Fig. R4 In vivo degradation of PPGs. **a** Degradation of PPGs with various crosslinking PDDA/pullulan mass ratios on Day 10 and Day 20 post implantation. **b,c** Optical images of Ce6@PPG (1:8 PDDA/pullulan mass ratio) samples, without (**b**) and with irradiation (**c**) at different time points post implantation. Irradiation: 640 nm, 5 mW cm⁻², 20 min on Day 0, 2, 4, and 6.

3. The hydrogel PPG was used for postsurgical prevention of tumor recurrence. Is it possible that this material be used for direct tumor inhibition?

Response: Thank the reviewer for the inspiring prospects on future applications of our hydrogel. The hydrogel PPG has achieved a promising therapeutic effect on the postsurgical prevention of tumor recurrence, and we would like to foresee the direct application of PPG for tumor inhibition. In addition, the PPG we provide in this paper is a drug delivery platform for general tumor treatment, including different tumor

models and different tumor treatment methods. We choose the postsurgical cancer treatment of 4T1 model, in that breast cancer has the leading incidence and mortality rates among females, and the long-term prognosis after surgery is unsatisfactory due to local recurrence and metastasis. Moreover, ICB therapy is especially unsuccessful in postsurgical treatment of breast cancer, attributed to its extremely immunosuppressive TME⁴⁻⁶. Therefore, we would like to challenge the postoperative 4T1 model using the outstanding antitumor immune response of our PPG platform. The successful immunostimulant postsurgical treatment of 4T1 mice implies that the ROS-responsive PPG hydrogel platform holds great promise as a generally applicable localized drug-delivery platform for sustained potentiation of ICB-based cancer immunotherapy. Applying PPG for direct tumor inhibition is currently under investigation.

4. The authors need to identify whether there is a photothermal effect of Ce6 in the strategy, in that PTT many also activate the immune system.

Response: Thank the reviewer for the very helpful suggestion. To verify the photothermal effect of Ce6, we have monitored the temperature changes of Ce6@PPG *in vitro* and *in vivo* with different irradiation time (Fig. R5). The temperature change of the hydrogel was less than 4 °C *in vitro* and 2 °C *in vivo* upon 30 min of irradiation, which proved that the PTT effect was minor.

Fig. R5 Temperature change of Ce6@PPG upon irradiation. **a** Temperature of Ce6@PPG *in vitro* and *in vivo* as a function of irradiation time. **b** Temperature change of Ce6@PPG *in vitro* and *in vivo* as a function of irradiation time. Irradiation: 640 nm, 5 mW cm⁻².

5. There are some typos in the manuscript. Please check through the manuscript to ensure they have been corrected.

Response: We appreciate the reviewer for the elaborate review and apologize for the typos. We have checked through the manuscript to ensure these typos have been corrected (highlighted in red in the revised manuscript).

Reviewer #3 (Remarks to the Author):

In this very interesting study by Zhang et al, the authors described a strategy of treating cancer, by combining immune checkpoint inhibitors and photodynamic therapy. They generated a ROS-responsive hydrogel PPG by crosslinking PDDA and pullulan, to realize sustained delivery of CD47-blocking antibodies and PS. ROS generated during PDT which could lead to degradation of antibodies reacted with PDDA and thus improved the efficacy of antibodies. This study focused on a very important topic to address critical questions in cancer immunotherapy. The manuscript was well-written and the data were well organized and presented. Despite the enthusiasm, several concerns need to be addressed to further improve the study.

1. There seems to be a concern regarding the dose of the functional antibodies can be released from the hydrogel. According to supplementary Fig 4, only less than 15% of loaded antibodies can be released in the in vitro assay, with or without LED irradiation. In addition, can the authors quantify the dose of loaded antibody that can be released in the in vivo setting?

Response: We really appreciate the reviewer for the enthusiastic and positive opinion about our work. We are also thankful for the valuable comments raised by the reviewer. For the in vitro experiment shown in Supplementary Fig. 4 (Supplementary Fig. 5 in the revised Supplementary Information), the reason why only less than 15% of the antibody could be released was that we only simulated the situation of PDDA degradation in the in vitro drug release study. In fact, both the oxidative degradation of PDDA and the metabolic degradation of pullulan contributed to the disintegration of the hydrogel and the release of the drugs. Since pullulan is a well-known polysaccharide with good biocompatibility and biodegradability, we only validated the effect of the oxidative degradation of PDDA on drug release in the in vitro study. Therefore, the gel degradation was slow under this condition, and the release of the antibody was insufficient. We agree with the reviewer that it is more important to quantify the dose of loaded antibody that can be released in the in vivo setting. We have conducted relevant experiments as the reviewer suggested to evaluate the in vivo antibody release. As shown in Fig. R6 (Fig. 3d and 3e in the revised manuscript), we used AF790-labeled goat anti-mouse IgG (AFIgG) as a surrogate of aCD47 to visualize the antibody release behavior by in vivo imaging system (IVIS). The formed AFIgG/Ce6@PPG hydrogels were implanted into the tumor resection cavities of mice in two random groups. When light irradiation was applied (640 nm, 5 mW cm⁻², 20 min on Day 0, 2, 4, and 6), the AFIgG was almost completely released within 21 days. As a comparison, it took about 28 days for the release of AFIgG to complete in the absence of light irradiation. We have addressed the effect of pullulan degradation on the PPG degradation and drug release on Page 9–10 in the revised manuscript (highlighted in red).

Fig. R6 In vivo antibody release behavior. **a** Fluorescence IVIS images depicting the in vivo retention of AFIgG fluorescence in mice with AFIgG solution (Free AFIgG), AFIgG/Ce6@PPG with (+L) or without (-L) the 640 nm LED irradiation. **b** Mean fluorescence intensity (MFI) of corresponding fluorescence IVIS images in (a). LED irradiation: 640 nm, 5 mW cm⁻², 20 min on Day 0, 2, 4, and 6. Data are mean ± SEM (n = 3 biologically independent samples).

2. Would co-incubation with Ce6 induce degradation of CD47 antibody?

Response: Thanks for the very good question. One of the most important features of PPG is its protection on the antibody. We have demonstrated in the manuscript that Ce6-generated ROS can trigger damage to protein (BSA), and PDDA was effective in protecting BSA from the damage (Supplementary Fig. 6 in the revised Supplementary Information). We have also showed that co-incubating aCD47 with Ce6 would affect the binding affinity of aCD47 to CD47 antigen using the Nano Surface Plasmon Resonance (NanoSPR) (Supplementary Fig. 7 in the revised Supplementary Information). PDDA could also preserve the activity of aCD47 by consuming the ROS generated by Ce6. In addition, to further examine Ce6-induced damage to aCD47, we used the NanoSPR to investigate the change of binding affinity of aCD47 antibody and CD47 antigen. As shown in Table R2, when co-incubating Ce6 with aCD47, the antibody-antigen association rate (K_a) and dissociation rate (K_d) of aCD47 remained unchanged in the absence of light irradiation. However, when irradiation was applied, the value of K_d greatly increased to $3.04 \times 10^{-4} \text{ s}^{-1}$, and the effective binding concentration (K_D) also increased over 4 times, clearly indicating that the activity of aCD47 was affected by the Ce6-generated ROS. Still, in the presence of PPG, both the values of K_d and K_D decreased accordingly, further evidencing the protective effect of PPG. We have added this study to the revised manuscript on Page 16 (highlighted in red) and the Supplementary Information (Supplementary Table 1, highlighted in red).

Samples	K_a ($M^{-1}s^{-1}$)	K_d (s^{-1})	K_D (M)
aCD47+L	8.32×10^4	8.47×10^{-5}	1.02×10^{-9}
aCD47+Ce6	4.87×10^4	5.30×10^{-5}	1.09×10^{-9}
aCD47+Ce6+L	6.61×10^4	3.04×10^{-4}	4.59×10^{-9}
aCD47+Ce6+PPG+L	2.31×10^4	5.79×10^{-5}	2.51×10^{-9}

Table R2 The binding affinity of aCD47 antibody and CD47 antigen measured using the NanoSPR technology under different conditions.

3. In the in vivo experiments shown in Fig4 and Fig5, the aCD47/Ce6 showed a strong effect on increasing CD8 and CD4 T cell infiltration but demonstrated no effects on inhibiting tumor growth. The authors need to comment on this.

Response: We really appreciate the reviewer for this valuable critique. We apologize for not stating clearly in the previous version of this manuscript. CD8⁺ and CD4⁺ T cells are important tumor infiltrating immune cells, and this analysis was conducted on Day 8 post implantation to make a comprehensive and timely comparison of the immune responses triggered by all treatments. Flow cytometry analysis (Fig. R7a and R7b, or Fig. 5b and 5c in the revised manuscript) showed that the proportions of CD8⁺ T cells and CD4⁺ T cells in “aCD47/Ce6+L” group were remarkably higher than those in “PBS” and “Blank PPG” groups. The results suggested that the mixed solution of aCD47 and Ce6, upon irradiation, induced immune response effectively in the early days, which was consistent with the tumor growth curve (Fig. R7c, or Fig. 4e in the revised manuscript). The tumor regrowth in “aCD47/Ce6+L” group was slow within the first two weeks. However, as the time proceeded, the tumor volume increased significantly. Since the solution of the free drugs diffused away gradually, the drug could not maintain an adequate concentration in the resection site for too long, and the therapeutic effect diminished accordingly. As a result, the mice in this group experienced a quick tumor recurrence starting from the third week post operation. We have added the additional discussion on Page 22 in the revised manuscript (highlighted in red) as the reviewer suggested.

Fig. R7 a, b Relative quantifications of CD8⁺ T cells (a) and CD4⁺ T cells (b) in tumors on Day 8 after the surgery. **c** Average tumor growth curves of different groups. Data are shown as mean \pm SEM ($n = 7-8$). The comparison of two groups was followed by student's t-test (two-tailed). * $P < 0.05$, ** $P < 0.01$, and *** $P < 0.001$.

4. In Fig5, M1 vs M2 macrophages were defined only based on CD80 and CD206 expression. Additional markers should be included to more precisely define M1 and M2 macrophages.

Response: We thank the reviewer for the very helpful suggestion. We apologize for not stating clearly in our previous manuscript. In addition to CD80 and CD206 expression, we also used F4/80 and CD11b markers to define M1 and M2 macrophages, as indicated in many previous literatures⁷⁻¹¹. M1 macrophages were labeled as CD80⁺CD11b⁺F4/80⁺ and M2 macrophages were labeled as CD206⁺CD11b⁺F4/80⁺ in the revised manuscript (highlighted in red). In addition, we have also added the gating strategies used for the flow cytometry analysis of M1 and M2, and other cells in the Supplementary Information (Supplementary Fig. 10 in the revised Supplementary Information, or Fig. R8 as shown below).

Fig. R8 Gating strategies used for cell sorting in flow cytometry analysis. **a** Gating strategy to sort CD8⁺ (CD3⁺CD8⁺) T cells and CD4⁺ (CD3⁺CD4⁺) T cells from Balb/c mice presented on Fig. 5a, 5b, and 5c. **b** Gating strategy to sort matured DC (CD11c⁺CD80⁺CD86⁺) cells from Balb/c mice presented on Fig. 5d. **c** Gating strategy to sort Treg (CD3⁺CD4⁺Foxp3⁺) cells from Balb/c mice presented on Supplementary Fig. 11. **d** Gating strategy to sort TEM (CD3⁺CD4⁺Foxp3⁺) cells from Balb/c mice presented on Fig. 6g. (E) Gating strategy to sort M1-like TAMs (F4/80⁺CD11c⁺CD80⁺) and M2-like TAMs (F4/80⁺CD11c⁺CD206⁺) from Balb/c mice presented on Fig. 5e, 5f and 5g.

5. In Fig5, the authors showed exciting efficacy of the aCD47/Ce6@PPG+L combination therapy. However, a direct comparison between the combination therapy and aCD47 alone is missing. In the two groups included in the experiments (aCD47/Ce6+L and aCD47/Ce6@PPG), it's possible that the efficacy of CD47 Ab was compromised due to ROS generated by Ce6. The author should directly compare the combination therapy (aCD47/Ce6@PPG+L) and aCD47 alone to demonstrate the efficacy of combination therapy is superior.

Response: Thank the reviewer for the very helpful comment. As the reviewer suggested, we conducted an additional in vivo experiments (Fig. R9) to make a direct comparison of the therapeutic effect of the combination therapy (aCD47/Ce6@PPG+L) and free aCD47. The dosage of aCD47 was set as 70 µg per mouse. The mice treated with aCD47 solution alone (the “aCD47” group) showed the similar tumor regrowth to those in the “PBS” group, suggesting that aCD47 alone had little effect on tumor inhibition. However, in the “aCD47/Ce6@PPG + L” group, tumor regrowth was effectively suppressed and the recurrence rates were significantly reduced, which was consistent with our previous animal experiments and demonstrated the superior therapeutic effect of the combination therapy. We have added this study to the revised manuscript (Page 18–19, highlighted in red) and the Supplementary Information (Supplementary Methods and Supplementary Fig. 9, highlighted in red).

Fig. R9 In vivo antitumor effect on 4T1-luc tumor-bearing BALB/c mice by PBS, aCD47, and aCD47/Ce6@PPG+L. **a** Average tumor growth curves of mice receiving different treatments, Data are shown as mean ± SEM (n = 8). The comparison of two

groups was followed by student's t-test (two-tailed). * $P < 0.05$, ** $P < 0.01$, and *** $P < 0.001$. **b** The photograph of recurrent tumors collected from the mice receiving different treatments on Day 34 post surgery.

6. More detailed description regarding the lung metastasis model should be provided. Mice numbers used for the in vivo experiments should be provided.

Response: Thank the reviewer for the suggestions. We have supplemented the description of establishing the lung metastasis model in the Methods section of the revised manuscript (Page 34 in the revised manuscript, highlighted in red). The added statement including the number of mice used for this experiment is as below: “On Day 60 post surgery, 3 remaining mice from each group were sacrificed and their lung tissues were harvested and fixed in Bouin's solution for metastasis analysis. The yellow nodules on the surface of lungs indicating the tumor metastasis sites were counted visually and examined by H&E staining”.

7. Experimental procedure of loading aCD47 and Ce6 to PPG should be included in the methods section with more details.

Response: Thank the reviewer for the helpful comment. We have supplemented more details about the preparation process of aCD47/Ce6@PPG in “Preparation of PPG” of the Methods section (Page 30 in the revised manuscript, highlighted in red). The added experimental procedure is “aCD47/Ce6@PPG was prepared immediately before use. 8 μL of Ce6 solution (10 mg mL^{-1}) and 7 μL of aCD47 solution (10 mg mL^{-1}) was added consequently to a lyophilized PPG to ensure each PPG was loaded with 80 μg of Ce6 and 70 μg of aCD47. The whole drug loading procedure was conducted in the dark”.

References

1. Tian, S. et al. Complete Degradation of a Conjugated Polymer into Green Upcycling Products by Sunlight in Air. *J. Am. Chem. Soc.* **143**, 10054-10058 (2021).
2. Ren, Y. L. et al. Effective Codelivery of lncRNA and pDNA by Pullulan-Based Nanovectors for Promising Therapy of Hepatocellular Carcinoma. *Adv. Funct. Mater.* **26**, 7314-7325 (2016).
3. Yuan, R. et al. Self-Assembled Nanoparticles of Glycyrrhetic Acid-Modified Pullulan as a Novel Carrier of Curcumin. *Molecules* **19**, 13305-13318 (2014).
4. Phuengkham, H., Song, C. & Lim, Y. T. A Designer Scaffold with Immune Nanoconverters for Reverting Immunosuppression and Enhancing Immune Checkpoint Blockade Therapy. *Adv. Mater.* **31**, 1903242 (2019).
5. Quail, D. F. & Joyce, J. A. Microenvironmental regulation of tumor progression and metastasis. *Nat. Med.* **19**, 1423-1437 (2013).
6. Ren, L. & Lim, Y. T. Degradation-Regulatable Architected Implantable Macroporous Scaffold for the Spatiotemporal Modulation of Immunosuppressive Microenvironment and Enhanced Combination Cancer Immunotherapy. *Adv. Funct. Mater.*

- Mater.* **28**, 1804490 (2018).
7. Chen, Q. et al. In Situ Sprayed Bioresponsive Immunotherapeutic Gel for Post-surgical Cancer Treatment. *Nat. Nanotechnol.* **14**, 89-97 (2019).
 8. Rao, L. et al. Hybrid Cellular Membrane Nanovesicles Amplify Macrophage Immune Responses Against Cancer Recurrence and Metastasis. *Nat. Commun.* **11**, 4909 (2020).
 9. Wang, C. et al. In Situ Formed Reactive Oxygen Species-responsive Scaffold with Gemcitabine and Checkpoint Inhibitor for Combination Therapy. *Sci. Transl. Med.* **10**, 429 (2018).
 10. Wei, B. et al. Polarization of Tumor-Associated Macrophages by Nanoparticle-Loaded Escherichia coli Combined with Immunogenic Cell Death for Cancer Immunotherapy. *Nano Lett.* **21**, 4231-4240 (2021).
 11. Wei, Z. et al. Huang, B.; Yang, X., Boosting anti-PD-1 therapy with metformin-loaded macrophage-derived microparticles. *Nat. Commun.* **12**, 440 (2021).

REVIEWER COMMENTS

Reviewer #2 (Remarks to the Author):

The authors have addressed all my questions. I recommend it to be accepted for publication without any further changes.

Reviewer #3 (Remarks to the Author):

The authors have addressed most of my concerns and the study has been improved. I still have two questions which I feel have not been fully addressed by revised manuscript:
First, can the authors explain why IgG was used as a surrogate of CD47 antibody when examining the antibody release? Have the authors confirmed that these two antibodies would have similar stability and biodistribution, and the data received from IgG can accurately reflect that of the CD47 antibody?
Second, the authors provided the gating strategy in FigR8 but it's quite confusing. Can the author provide more details about why distinct gating strategies were used for FSC and SSC gating in different experiments? And in FigR8a, large-sized cells with higher granularity (bigger FSC and SSC) were chosen for T cell analysis, but generally speaking, T cells are smaller than most other types of cells and thus this gating strategy may lead to a large group of T cells not being included in the analysis.

Response to Reviewer #3

The authors have addressed most of my concerns and the study has been improved. I still have two questions which I feel have not been fully addressed by revised manuscript:

First, can the authors explain why IgG was used as a surrogate of CD47 antibody when examining the antibody release? Have the authors confirmed that these two antibodies would have similar stability and biodistribution, and the data received from IgG can accurately reflect that of the CD47 antibody?

Response: We really appreciate the reviewer for the very valuable comments. As the reviewer pointed out, we used IgG as a surrogate of aCD47 antibody to evaluate the release of antibody from the gel in both in vitro and in vivo settings. One reason was that aCD47 antibody and fluorophore-labeled aCD47 antibody were expensive. In addition, according to its supplier, the isotype of the aCD47 antibody used in this study was rat IgG. Therefore, the physicochemical properties of both IgG and fluorophore-labeled IgG (AFIgG, commercially available), including molecular weight, surface charges, hydrophilicity, etc., were similar to those of the anti CD47 IgG. These properties are closely related to the release behavior of the antibody, so that the release behavior measured based on IgG should reflect that of the aCD47 antibody. The other reason why we used IgG in our release study was that PPG was constructed as a general platform for the sustainable localized release of antibodies. The release behavior of IgG we observed could be used to represent the release of IgG-isotype antibodies in general. We have added this discussion in the revised manuscript to clarify this point (Page 15, highlighted in red). There might be slight difference on the stability and biodistribution between IgG and aCD47, but we have confirmed the in vivo biofunction of the aCD47 antibody released from the gel as well as the antitumor immunotherapy effect. These results have validated that our PDDA-based hydrogel is a generally applicable localized drug-delivery platform for sustained potentiation of ICB-based cancer immunotherapy. We thank the reviewer for the critique, and we hope our explanation could address the concerns of the reviewer.

Second, the authors provided the gating strategy in FigR8 but it's quite confusing. Can the author provide more details about why distinct gating strategies were used for FSC and SSC gating in different experiments? And in FigR8a, large-sized cells with higher granularity (bigger FSC and SSC) were chosen for T cell analysis, but generally speaking, T cells are smaller than most other types of cells and thus this gating strategy may lead to a large group of T cells not being included in the analysis.

Response: We thank the reviewer for the thoughtful review and helpful comments. In our implementation, the lymphocytes derived from the tumor tissues or the lymph nodes of the same mouse sample were divided and stained by different fluorescence-labeled antibodies to measure different cell types (e.g., T cells, DC cells, etc.) separately. The different staining process could lead to different cell statuses. Therefore, the FSC and

SSC gating strategies of different cell types were different from each other. Nevertheless, the gating strategy of a specific cell type was kept consistent in different experiments throughout the study. Fig. R1 below (Fig. R8 in last response) showed the collection of representative gating strategies used for sorting each specific cell type. In addition, we really appreciate the reviewer for the question on the gating strategy for T cell analysis. As suggested by the reviewer, we have reanalyzed the flow cytometry data of T cells, with lower granularity chosen. The recounted percentages of tumor-infiltrating CD8⁺ and CD4⁺ T cells have slightly changed compared with the previous results (Fig. R2a). However, the trend of different treating groups remained the same, and the “aCD47/Ce6@PPG+L” group showed the highest CD8⁺ and CD4⁺ T cells percentages (Fig. R2b, 2c), demonstrating the exceptional immunostimulant effect by the synergistic combination of ROS and aCD47. We have updated the flow cytometry analysis results in the revised manuscript (Page 20-21 and Fig. 5a, 5b, 5c, highlighted in red) as well as the gating strategy in the Supplementary Information (Supplementary Fig. 10, highlighted in red).

Fig. R1 Gating strategies used for cell sorting in flow cytometry analysis. **a** Gating strategy to sort CD8⁺ (CD3⁺CD8⁺) T cells and CD4⁺ (CD3⁺CD4⁺) T cells from Balb/c mice presented on Fig. 5a, 5b, and 5c. **b** Gating strategy to sort matured DC (CD11c⁺CD80⁺CD86⁺) cells from Balb/c mice presented on Fig. 5d. **c** Gating strategy to sort Treg (CD3⁺CD4⁺Foxp3⁺) cells from Balb/c mice presented on Supplementary Fig. 11. **d** Gating strategy to sort TEM (CD3⁺CD8⁺CD44⁺CD62L⁻) cells from Balb/c mice presented on Fig. 6g. **e** Gating strategy to sort M1-like TAMs (F4/80⁺CD11c⁺CD80⁺) and M2-like TAMs (F4/80⁺CD11c⁺CD206⁺) from Balb/c mice presented on Fig. 5e, 5f and 5g.

Fig. R2 Flow cytometry analysis of CD4⁺ and CD8⁺ T cells. **a** Representative flow cytometry analysis of tumor-infiltrating CD4⁺ and CD8⁺ T cells gated on CD3⁺ T cells on Day 8 after the surgery. **b, c** Relative quantifications of CD8⁺ T cells (**b**) and CD4⁺ T cells (**c**) in tumors on Day 8 after the surgery.

REVIEWERS' COMMENTS

Reviewer #3 (Remarks to the Author):

My concerns have been adequately addressed. Thank you!